# Preliminary Broadband Dielectric Spectroscopy Insight into Compressed Orientationally Disordered Crystal-Forming Neopentyl Glycol (NPG)

**DOI:** 10.3390/ma18030635

**Published:** 2025-01-31

**Authors:** Aleksandra Drozd-Rzoska, Jakub Kalabiński, Sylwester J. Rzoska

**Affiliations:** Institute of High Pressure Physics, Polish Academy of Sciences, Sokołowska 29/37, 01-142 Warsaw, Poland; arzoska@unipress.waw.pl (A.D.-R.);

**Keywords:** neopentyl glycol, plastic crystals, orientationally disordered crystals, broadband dielectric spectroscopy, electric conductivity, dielectric constant, low-frequency changes, discontinuous phase transition, glassy dynamics, high-pressure barocaloric effect

## Abstract

This report presents the first results on broadband dielectric spectroscopy insights into ODIC-forming neopentyl glycol (NPG) under compression up to the GPa domain. Particular attention was paid to the strongly discontinuous phase transition: orientationally disordered crystal (ODIC)–solid crystal. The insights cover static, dynamic, and energy-related properties, namely evolutions of the dielectric constant, DC electric conductivity, and dissipation factor. Worth stressing are results regarding the pressure-related Mossotti catastrophe-type behavior of the dielectric constant, the novel approach to non-Barus dynamics, and the discussion on fundamentals of dissipation factor changes in NPG. The results presented in the given report also introduce new experimental evidence and model discussions regarding the nature of ODIC mesophase and discontinuous phase transitions. Notable is the significance of understanding the nature of the colossal barocaloric effect in NPG.

## 1. Introduction

Neopentyl glycol (NPG) has a wide range of applications, from the textiles, plastics, and coatings industries to pharmaceuticals and food processing [1,2]. It is also important for cognitive fundamentals. Namely, it can exist in an orientationally disordered crystal (ODIC) mesophase between liquid and solid crystal, matched with a discontinuous transition. ODIC-forming materials are in the plastic crystals family, with free orientational or rotational motions of molecules translationally ‘trapped’ in the crystalline network [3,4,5,6,7]. Two decades ago, the concept of the barocaloric effect, related to the ‘exploration’ of the discontinuous phase transition latent heat on compressing and decompressing, was introduced [8]. Only in 2019 was the colossal barocaloric effect (CBE) in NPG discovered, yielding hopes for groundbreaking applications [9,10]. For the phenomenon’s metric, the entropy change at the transition is used. For NPG, the colossal value $\Delta S\approx 400\;{JK}^{-1}{kg}^{-1}\;$was obtained when compressing up to~450 MPa [9].

Since then, NPG is considered the reference basis for new generations of thermal energy storage facilities, refrigerators, chillers, air-conditioners, and related technological solutions [11,12,13,14,15,16,17,18,19,20,21,22,23,24,25,26,27]. CBE-based devices can be wholly environment-friendly and more energy efficient than the omnipresent devices based on the circulation of special fluids and their almost continuous adiabatic decompression.

The barocaloric effect is related to hidden heat (L) coupled to discontinuous phase transition, explored during compressing and decompressing. The Clapeyron–Clausius (C-C) relation [27] constitutes the essential interpretative tool [9,14,15,16,19,22,23,24,25,28]:(1)$$\frac{dT_{m}}{dP}=T_{m}\frac{\Delta V}{L}=\frac{\Delta V}{\Delta S}$$
where $T_{m}$ denotes the melting temperature under a given pressure P; and ΔV, ΔS are for the volume and entropy changes at the discontinuous transition.

Fundamental BCE research, yielding essential support for applications and searches for new materials, follows the path set in discontinuous phase transition-related studies, particularly melting/freezing between liquid and solid crystal states [28,29,30,31,32,33]. The phenomenon has been intensively studied since the 19th century [33] but remains challenging. The cognitive situation contrasts continuous, critical phase transitions with the grand universalistic success of *Critical Phenomena Physics* [34,35,36,37].

To a large extent, cognitive problems in the ultimate explanation of discontinuous phase transitions can be related to the apparent absence of long-range pretransitional effects [28,29,30,31,32,33]. Generally, fundamental studies on the melting/freezing transition CBE phenomenon are focused on magnitudes in C-C Equation (1) and the comparison of structural and spectroscopic properties in adjacent phases [9,10,11,12,13,14,15,16,17,18,19,20,21,22,23,24,25,26,27,28,29,30,31,32,33]. A hope for a cognitive breakthrough is suggested in recent findings of critical-like premelting effects appearing when using dielectric methods [38,39,40,41]. The second cognitive problem is the still limited evidence for high-pressure studies [28,29,30,31,32,33]. It is notable that the high-temperature ODIC phase can be considered a specific model liquid, where the ‘freedom’ is related solely to orientations. Hence, broadband dielectric spectroscopy (BDS) [42], inherently coupled to electric field action, is the primary method for testing the ODIC mesophase [3,4,5,6,43].

When commenting on the importance of high pressure and dielectric studies for NPG, worth noting is the recent model analyses of entropy changes ΔS for discontinuous phase transitions leading to the following output relation [44]:(2)$$\Delta S={\Delta s}_{1}+{\Delta s}_{2}+{\Delta s}_{3}=\int_{T_{ref.}}^{T}\frac{C_{P,E}}{T}dT+\int_{P_{ref.}}^{P}v\alpha dP+\varepsilon_{0}\int_{0}^{E}v{\left(\frac{d\varepsilon }{dE}\right)}_{P}EdE$$
where $C_{P,E}$ is the heat capacity related to a constant pressure and/or electric field, v is the specific volume per unit mass, and α is the isobaric thermal expansion concerning subsequent pressures.

The first term ${\Delta s}_{1}$, can be related to the general internal energy change. The second term ${\Delta s}_{2}$ can be retrieved as follows at the transition; it indicates the significance of the compressibility change at the ${\Delta \chi }_{T}$ discontinuous transition and the pressure dependence of the phase transition, namely
$\alpha =\left(1/V\right){\left(\partial V/\partial T\right)}_{P\;}={\left(\partial V\partial P/\partial T\partial P\right)}_{T_{m,}P_{m}}=\left(1/V\right){\left(\partial V/\partial P\right)}_{T_{m,}P_{m}}{\left(dT_{m}/dP\right)}^{-1}\Rightarrow \alpha ={\Delta \chi }_{T}{\left(dT_{m}/dP\right)}^{1}$. It explicitly shows the importance of the compressibility difference for adjacent phases. It is particularly impressive for NPG, due to the ductility of the high-temperature plastic crystal ODIC mesophase and the ‘hardness’ of the solid-state crystal. Also notable is the significance of the discontinuous phase transition $T_{m}\left(P\right)$ pattern. The last term in Equation (2), ${\Delta s}_{3}$, directly recalls the significance of dielectric constant change when passing the transition.

This report shows the first multi-aspect broadband dielectric spectroscopy insights for neopentyl glycol (NPG) on compression in the broad surrounding of the discontinuous phase transition between the ‘soft’ ODIC phase and the ‘hard’ crystal phase. The report introduces new evidence regarding the ODIC mesophase’s properties, the ODIC–crystal or even melting/freezing discontinuous phase transition, and the colossal barocaloric effect in NPG and related materials [45,46].

## 2. Temperature-Related BDS Studies Under Atmospheric Pressure in Neopentyl Glycol

There is a large amount of experimental evidence regarding broadband dielectric spectroscopy in ODIC-forming materials. It focuses on the complex, previtreous dynamics, since such materials often can be supercooled down to the amorphous glass state ([42,43] and (refs. therein)). They are motivated by the glass transition grand challenge, considered an essential problem for 21st-century physics and material engineering [47,48,49,50,51]. Glass transition studies are essentially focused on the previtreous domain and the evolution of the primary relaxation time and, eventually, its distribution, i.e., properties associated with the primary loss curve in the imaginary part of dielectric permittivity $\varepsilon^{\prime \prime }\left(f\right)$ detected in BDS experiments. Notable is the cognitive differences between temperature and pressure studies. The first one is related to activation energy changes, whereas compressing shifts the activation volume. Such pressure studies are still very limited for ODIC-forming materials, and the first results were reported only two decades ago [52,53].

As for NPG, Tamarit et al. [54] presented changes in the primary relaxation time in the ODIC phase of NPG for temperatures ranging from ~353 K to 305 K, i.e., covering ca. 50% of the ODIC phase range in this material. Although the functional parameterization is not discussed, the presented slightly nonlinear pattern in the Arrhenius scale plot $\ln\tau \left(T\right)$ vs. 1/T indicates possible super-Arrhenius (SA) dynamics, considered the universalistic feature for previtreous dynamics. In subsequent reports, for similar ODIC-forming materials, the parameterization via the Vogel–Fulcher–Tamman (VFT) equation, i.e., the replacement equation for the general SA relation, is shown [55,56]:(3)$$\tau \left(T\right)=\tau_{\infty }exp\left(\frac{E_{a}\left(T\right)}{RT}\right)\Rightarrow \tau \left(T\right)=\tau_{\infty }exp\left[\frac{D_{T}T_{0}}{T-T_{0}}\right]$$
where the left part is for the general SA equation, with the apparent (temperature-dependent) activation energy $E_{a}\left(T\right).$ It reduces to the basic Arrhenius pattern for $E_{a}\left(T\right)=E_{a}=const$. The right part is for the VFT replacement equation. Equation (1) is for the supercooled liquid-like temperature domain $T>T_{g}$; $T_{0}<T_{g}$ is the extrapolated VFT singular temperature; and $T_{g}$ is the glass temperature, which can be estimated via the empirical condition *τ*$\left(T_{g}\right)=100\;s$. $D_{T}$ is the fragility strength parameter; $D_{T}T_{0}=const$ [42,50].

The VFT equation is the commonly used dependence for describing previtreous dynamics, including the vitrifying ODIC phase. Notwithstanding, since 2006, the prevalence of the critical-like parameterization in the ODIC phase has been presented [50,57]:(4)$$\tau \left(T\right)=\tau_{0}{\left(T-T_{C}\right)}^{-\varphi }$$
where $T_{C}<T_{g}$ is the extrapolated singular temperature, and the exponent φ=9–15 for different ODIC-forming materials.

The primary relaxation time-focused BDS studies in NPG constitute a particular experimental challenge, since they require multi-GHz range measurements in a relatively volatile and sensitive to contaminations material. The subsequent report for for NPG appeared in 2021 [58]. It was related to frequencies f≤1 MHz and covered liquid, ODIC, and crystal phases in the temperature range from 416 K to 293 K. It focused on the evolution of DC electric conductivity, for which the VFT-type portrayal (Equation (1)) was suggested [58]:(5)$$\sigma \left(T\right)=\frac{A}{\sqrt{T}}exp\left(\frac{-B}{T-T_{V}}\right)$$
where A,B=const, and $T_{V}$ is the extrapolated singular temperature.

The description via the above relation was evidenced in the liquid phase for the range covering ~10 K and in the ODIC phase (denoted as Phase I) in the domain covering ~60 K, namely starting 22 K below the melting temperature and terminating ~7 K before the transition to the solid crystal phase. For the latter, the basic Arrhenius dynamics ($E_{a}^{\sigma }=const$) are reported. In ref. [56], also, spectra of imaginary parts of the dielectric modulus $M^{\prime \prime }\left(f\right)$ and electric impedance $Z^{\prime \prime }\left(f\right)$ for four temperatures were superposed and preliminarily discussed. The authors of ref. [58] heuristically suggested the importance of the proton hoping mechanism, which is influenced by the rotational freedom of molecules and intrinsic defects, for translational processes.

Very recently, BDS studies extend from the low-frequency and mid-frequency domain to a high-frequency, explicitly orientation process-focused region [43]. As for applications recalled in the section above, new findings related to constant dielectric changes are particularly noteworthy. First, the unique pattern for the dielectric constant or more precisely, dielectric permittivity χ=ε−1 has been shown [43]:(6)$$\chi \left(T\right)=\varepsilon \left(T\right)-1=\frac{A}{T-T^{*}}\Rightarrow \chi^{-1}\left(T\right)=A^{-1}T-A^{-1}T^{*}$$
where $T^{*}$ is the singular, ‘critical’, temperature related to $\chi^{-1}\left(T^{*}\right)=0$, the amplitude A=const, and then $A^{-1}T^{*}=const$.

Such a pattern was noted earlier for the liquid and ODIC phases of cyclooctanol, which can suggest its universality [43]. A model explanation of such behavior was proposed. It suggested obeying the Clausius–Mossotti local field, directly leading to Mossotti catastrophe-type behavior [58,59,60,61,62,63], correlated with Equation (6). Such behavior is formally forbidden for dielectric materials, except the paraelectric phase of ferroelectric materials. Nevertheless, the translational confinement of molecules in crystal lattices and the orientational freedom of the permanent dipole moments conjugated to them lead to the formal fulfillment of the essential condition for the Clausius–Mossotti model of the local field in dielectric materials, namely the presence of non-interacting or very weakly interacting permanent dipole moments [43,59,60,61,62,63,64].

For dynamics, tested via the evolution of DC electric conductivity, the explicit prevalence of the ‘activated&critical’ model relation, introduced by Drozd-Rzoska [65], was shown [43]:(7)$$\sigma^{-1}\left(T\right)=C_{\Gamma }{\left(\frac{T-T^{+}}{T}\right)}^{-\Gamma }{\left[\exp\left(\frac{T-T^{+}}{T}\right)\right]}^{\Gamma }=C_{\Gamma }{\left(t^{-1}\exp t\right)}^{\Gamma }$$
where $t=\left(T-T^{+}\right)/T$, $T^{+}$ is the extrapolated singular temperature, the prefactor $C_{\Gamma }=const$, and the exponent Γ=const.

The values of the exponents are in the range 4<Γ<25, and for its smaller values, the ‘critical-like’ term dominates, leading to the approximation by Equation (4) [65].

The distortions-sensitive validation of the above relation is the simple, quasi-critical pattern for apparent activation enthalpy $H_{a}\left(T\right)$ changes [43,65]:(8)$${H_{a}\left(T\right)=\left[\frac{dln\sigma^{-1}\left(T\right)}{d\left(1/T\right)}\right]}^{-1}=HT+HT^{+}\Rightarrow \frac{dln\sigma^{-1}\left(T\right)}{d\left(1/T\right)}=\frac{H}{T-T^{+}}$$
where H=const, and $T^{+}$ is the extrapolated singular temperature related to the ${\left[H_{a}\left(T^{+}\right)\right]}^{-1}=0$, $HT^{+}=const$ condition. $H_{a}\left(T\right)$ can also be considered the steepness index for the Arrhenius scale presentation metric; it is also directly related to so-called apparent fragility, the normalized metric of non-Arrhenius behavior introduced for ‘glassy’ dynamics.

Notably, slightly earlier the above patterns moted in the ODIC phase of cyclooctanol [66], which leads to the question of their universality.

## 3. Materials and Methods

Neopentyl glycol, with the highest available purity, was purchased from Sigma-Aldrich (Saint Louis, MO, USA) NPG Glycol (CAS 126-30-7). It shows the following phase sequence under atmospheric pressure: $Liquid\to \left(T_{L-O}=403.6K\right)\to ODIC\to \left(T_{O-C}=315.6K\right)\to Solid\;Crystal$.

High-pressure measurements were carried out using two setups. The first one was used to study up to 700 MPa. It is shown in Figure 1. For this facility, the pressure pump supports the pressure chamber, where Plexol is used to transmit the pressure. It consists of a pressure chamber linked to the pressure pump. The pressure systems were technically designed and constructed by Unipress Equipment (Warsaw, Poland). The pressure pump system consisted of two complementary pumps. The first one, with the large volume of the liquid transmitting pressure, enabled compression up to even 900 MPa, with ~2 MPa steps. The switch to the ‘micro-pump’, at any established pressure, enables ‘subtle’ pressure changes in the range of 200 MPa, with the resolution ~0.02 MPa.

The chamber was surrounded by a jacket, linked to a large-volume (V=25 L) thermostat with external circulation. Both temperature and pressure were computer-controlled. Temperature was measured by the copper-constantan thermocouple placed in the pressure chamber and two Pt100 resistors along the chamber to monitor a possible temperature gradient. The pressure was monitored by a tensometric meter, with ±0.1 MPa precision. Tested samples were placed in the measurement flat parallel capacitor in the pressure chamber. The pressure was transmitted to the sample inside the capacitor via the deformation of an elastic element. Its design is shown in ref. [50]. The measurement capacitor was made from Invar, with a d=0.3 mm gap between capacitor plates.

The second setup focused on extending the pressure up to challenging 2 GPa. It consisted of a pressure chamber with an internal diameter of 12 mm. The capacitor with the tested sample was placed in the Teflon tube with an external diameter only slightly lower than the diameter of the chamber. Inside the Teflon tube, a flat parallel measuring capacitor was created on the basis of an Invar cylinder cut along the long axis. The distance between the plates was 0.3 mm.

High and very high pressures acting on the tested sample were created by the piston being pressed into the pressure chamber, causing the Teflon tube’s plastic (reversible) deformation. The pressure chamber was also surrounded by a jacket, allowing liquid circulation from a large-volume thermostat with external circulation. The temperature was measured using a thermocouple inside the chamber. Changing the experimental pressure technique from method 1 to method 2 allowed us to avoid problems with a strong increase in the viscosity of the medium (liquid) transmitting the pressure in method 1. The thin tube caused the immediate transmission of pressure to the sample from pressurized transmission liquids, as well as avoiding contamination. The usage of Plexol, a liquid specially ‘designed’ for high pressure, guaranteed the same and not high viscosity during processing, which allowed the lack of its impact on the measurement process and on the destruction of the (expensive) pressure pump when approaching the GPa domain.

Method 2 is associated with limitations in high-pressure value estimations to at least ±5 MPa. It can be calculated from the force acting on the piston, creating pressure by high-pressure press. It has to be calibrated by a testing pressure induced using the manganin coil electric resistance placed in the chamber. Consequently, applying the second method for pressures above a few hundred MPa can be advised.

BDS measurements were carried out using the Alpha Novocontrol (Montabaur, Germany) BDS spectrometer, supported by Novocontrol system software, allowing the avoidance of parasitic capacitances and yielding directly the real and imaginary parts of dielectric permittivity or electric conductivity as the output of measurements. The analyzer enables 5- or 6-digit resolution scans. Tests were carried out for the U=1 V measurement voltage, which guaranteed the optimal measurement resolution. BDS studies under pressure are still limited to a few MHz in frequency because of unsolved technical challenges. For limited cases, it is 10 MHz, but most often, 1–3 MHz [53,62,65]. The terminal pressure depends on the capacitance–resistivity–frequency chart for the given impedance analyzer with respect to the tested sample parameters. Parasitic capacitances were removed following the protocol of the impedance analyzer producer. Addionally, tests using menthol and nitrobenzene as standard liquids with well-known dielectric parameters were used for the validation.

The reproducibility of the results in the BDS measure in the under-GPa (Method 1) and above-GPa domain depends on the precision of pressure estimation for both methods, as noted above. Nevertheless, the relative error ΔP/P does not differ strongly, when taking into account rising pressure values. The analyzer enables the cumulative analysis of the registered response. Additionally, the results presented are based on three experimental sets; no significant value differences were detected. For the temperature stabilization, it is always very high for properly designed set-ups. The temperature was scanned within and along the chamber to avoid gradients. Compression was performed in small pressure steps, and a 10 min pause occurred before the subsequent measurements. Hence, it could not change the temperature of the tested sample.

The example of experimental spectra detected in BDS-based scans is presented in Figure 2 and Figure 3 for the real imaginary parts of dielectric permittivity: $\varepsilon^{*}=\varepsilon^{\prime }+i\varepsilon^{\prime \prime }$, where $\varepsilon^{\prime }=C/C_{0}$ and $\varepsilon^{\prime \prime }=1/\omega CR$, $C\left(f\right)$ and $C_{0}$ are detected electric capacitances for the capacitor with the sample and ‘empty’, $R\left(f\right)$ is the resistivity, and ω=2πf [42] They were the base for deriving the properties discussed in the subsequent section. In Figure 2, significant frequency domains are indicated.

## 4. Results and Discussion

### 4.1. Real Part of Dielectric Permittivity, Moderate Pressure

The dielectric constant is the first, and still essential, characterization of dielectric materials, introduced by Michel Faraday [67]. It is related to the horizontal ‘static’ domain of the real part of dielectric permittivity $\varepsilon^{\prime }\left(f\right)$. For NPG, the static domain appears explicitly only in the ODIC phase, for ~2 kHz<f<~2 MHz, as shown in Figure 2 by the horizontal line in gray. For the solid crystal phase, the extent of the static domain is qualitatively lesser. For frequencies above the static domain, the relaxation domain with the primary loss curve directly addresses the reorientations of permanent dipole moments. It was not possible to access these frequencies because of the technical limitations of the frequency range in high-pressure studies [52,53,65].

Below the static domain, the boost in $\varepsilon^{\prime }\left(f\right)$
and
$\varepsilon^{\prime \prime }\left(f\right)$
values is linked to the low-frequency (LF) domain. Generally, it is related to ionic species in dielectric materials, which can contribute to dielectric materials for low enough frequencies.

The strong rise of dielectric permittivity is essentially linked to generally recalled ‘ionic species’ or ‘contaminations’ [68,69,70,71,72,73,74,75]. They can also be explained by translational shift, with respect to the average position, for basic molecules of a given system/material [50].

Worth recalling is the still-existing challenge for dielectric permittivity parameterization in the LF domain, both for $\varepsilon^{\prime }(f)$ and $\varepsilon^{\prime \prime }(f)$ [66,67,68,69,70,71,72,73]. It is notable that the latter is directly related to electric conductivity, as indicated above.

Figure 4
shows an empirical attempt to overcome this problem. It presents the changes of the real part of the dielectric permittivity reciprocal, revealing features not explicitly visible in
Figure 2. Horizontal lines in gray indicate the static domain, where frequency shifts do not yield changes in $\varepsilon^{\prime }\left(f\right)$ values. A simple linear pattern emerges for the LF domain, i.e., frequencies below the static domain. It leads to the following set of equations:
(9)$$1/ln\varepsilon^{\prime }\left(f\right)=a+lnf\Rightarrow ln\varepsilon^{\prime }\left(f\right)=1/\left(a+lnf\right)\Rightarrow \varepsilon^{\prime }\left(f\right)=exp{\left(a+lnf\right)}^{-1}$$
where a, b are empirical constants.

Figure 5 presents isothermal pressure changes in the dielectric constant in the surroundings of the discontinuous phase transition. The plot is in the semi-log scale to support the insights both in the ODIC and crystal phase, despite strong different values. The impact of frequency shift to the LF domain is visible in both phases. For the static domain, in the frequency range ~100 kHz, changes of $\varepsilon \left(P\right)$ seem to be nearly linear. Visible is a weak pretransitional rise in the solid phase for pressures just above the phase transition.

Studies of temperature changes in the dielectric constant in dipolar liquid dielectrics showed that the general tendency in $\varepsilon \left(T\right)$ evolution can indicate the dominant way of permanent dipole arrangements. Namely, for $d\varepsilon \left(T\right)/dT>0$ a preference for their antiparallel arrangement can be expected, and for $d\varepsilon \left(T\right)/dT<0$ the parallel one [61]. A similar ‘rule’ should be expected in the ODIC phase, with a large range of permanent dipole moments’ orientational freedom. Implementing this reasoning to isothermal changes on compression, one should expect $d\varepsilon \left(P\right)/dP<0$ as the indicator of the preferable antiparallel arrangement and $d\varepsilon \left(P\right)/dP>0$ for the parallel one.

### 4.2. Electric Conductivity, Moderate Pressure

Figure 6 displays the results of the transformation of the experimental data given in Figure 3 to the electric conductivity representation, $\sigma^{\prime }=\varepsilon_{0}{\omega \varepsilon }^{\prime \prime }\left(f\right)$, ω=2πf [61], to enable an insight into dynamic properties overcoming the mentioned frequency range limitations in high-pressure studies. The gray horizontal line indicates the DC electric conductivity domain, which can be considered the ‘static domain’ parallel for dynamic properties. For this domain, the frequency shift does not change the electric conductivity value: $\sigma^{\prime }\left(f\right)=\sigma =\sigma_{DC}$. In the ODIC phase, the DC domain covers three decades in frequency: 1 Hz<f<1 kHz. In the solid crystal phase, such behavior emerges only for f→1 Hz. Pressure changes in electric conductivity, including the DC domain, are presented in Figure 7. Notable is the complex pattern in the solid crystal phase.

Vertical continuous arrows indicate possible phase transitions or transformations in this domain. The first one is for
$P_{1}\approx 510\;MPa$. The second one is more complex, since its hallmarks are frequency-dependent:
$P_{2}\approx 415-450\;MPa$.

The ODIC–crystal discontinuous transition is explicitly visible at
$P_{O-C}=148\;MPa$, as shown by the ‘thick’, dashed arrow.

The striking feature is the pretransitional behavior in the solid crystal phase, which extends up to
$P_{O-C}+250\;MPa$. At least for the first 150 MPa from the transition, it can be approximated by the following critical-like relation:
(10)$$\sigma \left(P\right)=\sigma_{C}+A{\left(P-P^{*}\right)}^{\phi =1/2}+a\left(P-P^{*}\right)$$
where
$\sigma_{C},A,a\;$
are constant parameters, the exponent
ϕ=1/2, and the singular pressure
$P^{*}\approx 144\;MPa$. Note—it is close to the discontinuous transition
$P_{O-C}=148\;MPa$.

The above result is for the DC limited, i.e.,
f=1 Hz. The pretransitional effect diminishes when the frequency is increased, i.e., it is shifted away from the DC domain limit.

The results in Figure 6 are presented in the ‘Barus scale’ i.e., $ln\sigma^{\prime }\left(P\right)$ vs. P, which can be considered as the pressure counterpart of the Arrhenius scale $ln\sigma^{\prime }\left(T\right)$ vs. 1/T [42] used in temperature studies under atmospheric pressure. For the ‘Barus scale’, the linear dependence validates the simple behavior with the constant activation volume $V_{a}=const$ in the given pressure range, namely:(11)$$\sigma \left(P\right)=\sigma_{0}exp\left(cP\right)\Rightarrow \sigma \left(P\right)=\sigma_{0}exp\left(P\frac{V_{a}}{RT}\right),\;T=const$$ where the left side is for the original Barus report [76] and the right side with the activation volume $V_{a}$; in the given case $V_{a}=const$ and the empirical Barus parameter c=const; R means the gas constant.

It is notable that the authors of the given report in refs. [50,65] introduced the name Barus scale to honor the author of ref. [76], who introduced pressure portrayal via the left side of Equation (11). It also included the names super-Barus behavior and super-Barus (SB) equation, with the pressure-dependent apparent activation volume ${\;V}_{a}\left(P\right)$ [50,65,77].

### 4.3. Dielectric Loss Factor Changes

The dissipation factor D addresses both dynamics and energy-related issues in dielectric materials. It is defined as follows [61,64,78,79,80]:(12)$$D=tan\delta =\frac{\varepsilon^{\prime \prime }}{\varepsilon^{\prime }}=\frac{energy\;loss}{energy\;stored}\left(per\;cycle\right)$$

It is the crucial parameter in the evaluation of materials used for isolation in electric equipment, from the solid covers of cables, capacitors, and some machines to transformers’ oil. Its measurement can indicate the presence of moisture, aging, or other degradation factors. Such tests are essential to predict dielectric materials’ life expectancy and maintenance in tested elements [78,79,80]. Surprisingly, this magnitude, commonly used in material engineering, is hardly discussed in fundamental studies of dielectric properties in solids and liquids [42,60,61,62,63,64].

Figure 8 shows pressure changes of the dissipation factor in neopentyl glycol (NPG), derived from experimental data shown in Figure 2 and Figure 3. The linear behavior in the semi-log scale plot 8 suggests the evolution following the parallel of the Barus relation, namely: $tg\delta \left(P\right)=D_{0}exp\left(cP\right)$, c=const. Nevertheless, pretransitional effects are visible in the solid phase, particularly for the lowest presented frequency. Notable is the significant change in values from $tg\delta \left(P\right)\sim 0.01$ for f=100 kHz, which in applications would be classified as an excellent isolator (no energy loss), to $tg\delta \left(P\right)\sim 100$ for f=10 Hz, i.e., the value linked to an extremely poor isolator and substantial energy losses that can be converted to heat.

Figure 9 supplements the above discussion by showing the $tan\delta \left(f\right)$ frequency-related spectrum. In the ODIC phase, the plot suggests the evolution $tan\left(f\right)\propto f^{-1}$ for 30 Hz<f<300 kHz. It is considered a pattern for a non-ideal capacitor in an AC circuit when the ratio of resistive to reactive power loss is considered [78,79,80]. When commenting on such behavior for NPG, one can recall Figure 2 and Figure 3, where in the ODIC phase $\varepsilon^{\prime }\left(f\right)=\varepsilon \approx const$ in the static domain, and $\varepsilon^{\prime \prime }\left(f\right)\propto f^{-1}$ for low frequencies. The latter is related to the DC electric conductivity domain in Figure 6.

Although the ‘canonic’ static domain terminates at f~10 kHz, it remains only slightly distorted down to 50–100 Hz in the ODIC phase. It is notable that usually, in molecular liquid dielectrics, f~1 kHz is the definitive onset of the LF domain boost of dielectric permittivity. With these features of spectra presented in Figure 2, Figure 3 and Figure 6, confronted with the definition of the dissipation factor (Equation (12)), one obtains $\varepsilon^{\prime }\;\left(f\right)\approx const$, even down to ~100 Hz. In the same domain, the DC electric conductivity horizontal evolution is equivalent to $\varepsilon^{\prime \prime }\left(f\right)\propto f^{-1}$. Hence, for the ODIC phase in NPG, the appearing model-relation $tg\delta \left(f\right)\sim f^{-1}$ can be linked to the dominance of translational processes.

The crossover
$d{log}_{10}\left(tan\delta \right)/d{log}_{10}>0\leftarrow d{log}_{10}\left(tan\delta \right)/d{log}_{10}<0\;$
for low frequencies is associated with the definitive leave of
$\varepsilon^{\prime }\left(f\right)$
near the static domain and the LF domain’s substantial boost in values.

### 4.4. Dielectric Constant: Extreme Pressure Tests

The above evidence suggests linear changes in the dielectric constant and simple Barus-type evolution for electric conductivity. However, it can also be linked to the relatively small pressure range:
$\Delta P\left(ODIC\right)\sim 150\;MPa$. To test this hypothesis, challenging studies for isotherms reaching
T=110 °C, where the ODIC phase extends up to
P=1.2 GPa, were carried out.

The mentioned extreme pressure tests reveal a slight distortion from the linear behavior in
$\varepsilon \left(P\right)$
changes.
Figure 10
presents the results of these data via the analysis of the dielectric susceptibility
χ=ε−1 reciprocal. The explicit linear behavior suggests the following scaling pattern:
(13)$$\varepsilon \left(P\right)-1=\chi \left(P\right)=\frac{A}{P^{+}-P}\Rightarrow \chi^{-1}\left(P\right)=a+bP$$ where
A=const; $P^{+}$ is the extrapolated singular pressure;
$a=A^{-1}P^{+}$; and
$b=-A^{-1}P$.

Notable is the similarity of the pressure-related Equation (13) and temperature-related Equation (6) The latter has been just reported in the ODIC phase of NPG [43] and earlier in cyclooctanol [66]. In those reports, it was explained via precursors of Mossotti catastrophe behavior and the validation of Clausius–Mossotti simple local field modeling. The similarity of Equation (6) for temperature and Equation (13) for pressure changes in the ODIC phase also recalls the Isomorphism Postulate within *Critical Phenomena Physics* [35], suggesting the same critical scaling pattern when a given physical property is tested along two different field variable related paths of approaching continuous phase transition. It is particularly related to values of universal critical exponents for pressure and temperature paths of approaching the critical. For the ODIC phase of NPG, Equation (13) for $\chi \left(P\right)$ and Equation (6) $\chi \left(T\right)$ also resemble the Curie–Weiss-type behavior pattern in the paraelectric phase of ferroelectric materials, coupled to the susceptibility (compressibility) exponent (γ=1).

### 4.5. DC Electric Conductivity: Extreme Pressure Test

Figure 11
presents the evolution of DC electric conductivity in the ODIC phase extended up to
P=1.2 GPa
‘extreme’ pressure. It reveals a slight distortion from the apparent Barus pattern visible in
Figure 7, where the ODIC phase covers only
~0.15 GPa. The nonlinear, non-Barus, behavior in
Figure 11
is visible but too weak for a reliable and decisive fitting by a model equation.

To overcome this problem, the analysis recalling the super-Barus equation, i.e., the extension of Equation (11) with the pressure-dependent apparent activation energy $V_{a}\left(P\right),$ can be considered. Generally, such a super-Barus equation cannot be used for fitting electric conductivity $\sigma \left(P\right)$, primary relaxation time $\tau \left(P\right),$ or viscosity $\eta \left(P\right)$ data because of the unknown general pattern of $V_{a}\left(P\right)$ changes. Nevertheless, it is commonly used for estimating the apparent activation via the following relation Equation (11) [81,82,83,84,85,86,87,88,89,90]:(14)$$ln\sigma \left(P\right)=ln\sigma_{0}+P\frac{V_{a}}{RT}\Rightarrow RT\frac{dln\sigma \left(P\right)}{dP}=V_{\#}\left(P\right)$$

In the analysis using the above equation, it is assumed that the right part of the above relation estimates the apparent activation volume estimation, i.e., $V_{\#}\left(P\right){=V}_{a}\left(P\right)$. Unfortunately, this assumption is essentially wrong, i.e., $V_{\#}\left(P\right)\neq V_{a}\left(P\right),$ and $V_{\#}\left(P\right)\sim dln\sigma \left(P\right)/dP$ determines the steepness index for the non-Barus type experimental data presented in the Barus scale plot [50,77]. Namely, the right part of Equation (14) can be obtained only by assuming $V_{a}=const$. The proper analysis of the super-Barus relation gives [50,77].(15)$$ln\sigma \left(P\right)=ln\sigma_{0}-P\frac{V_{a}\left(P\right)}{RT}\Rightarrow RT\frac{dln\sigma \left(P\right)}{dP}=V_{a}\left(P\right)-\frac{dV_{a}\left(P\right)}{dP}$$

The term
$dV_{a}\left(P\right)/dP=0$
only for the basic Barus equation with
$V_{a}\left(P\right)=V_{a}=const$. A protocol overcoming this problem and allowing the determination of the real apparent activation volumes for subsequent pressures was proposed by Drozd-Rzoska [77], where the prevalence of the following extension of the SB relation was also indicated:(16)$$\sigma \left(P\right)=\sigma_{0}exp\left(-\Delta P\frac{V_{a}\left(P\right)}{RT}\right)$$ where
$\Delta P=P-P_{Sp}$; $P_{Sp}<0$
is the absolute stability (spinodal) limit for the tested isotherm
T=const.

Originally in refs. [50,77], the discussion was carried out in frames of the primary relaxation time. Equation (16) takes into account that for real liquids and solids, the pressure P=0 does not constitute any specific reference. Such a value can be passed without any hallmark into the negative pressure domain, where isotropic stretching until reaching $P_{Sp}<0$ value is possible. P=0 is the terminal point only for gases [50,77].

Figure 12 presents pressure changes of the apparent activation volume, related to Equation (16) and recalling the analysis presented in ref. [77]. It is notable that for the basic Barus relation, i.e., Equation (11) with $V_{a}=const$, the horizontal line related to $V_{a}=const$ should appear.

The linear parameterization appearing in Figure 12 can be substituted into Equation (16), yielding the following dependence portraying experimental data in the ODIC phase of NPG:
(17)$$\sigma \left(P\right)=\sigma_{0}exp\left(-\Delta P\frac{a+bP}{RT}\right)=\sigma_{0}exp\left(-\Delta P\frac{a+bP}{RT}\right)$$ where
$\Delta P=P-P_{Sp}$, and values of $P_{Sp}$, a,b
are given in the caption of
Figure 12.

The result of such a description is shown by the red curve portraying experimental data in Figure 12: Notably, the described routine avoids a nonlinear fitting.

### 4.6. Pressure Dependence of the ODIC–Crystal Discontinuous Phase Transition Temperature

Figure 13 concludes the discussion of NPG pressure-related features presented in the given report by showing the pressure dependence of the discontinuous ODIC–crystal phase transition. Data were determined by detecting dielectric constant step changes when passing the transition.

Results presented in Figure 13 can be parameterized by the Simon–Glatzel equation [50,91], as shown by shown by the solid curve in the plot:(18)$$T_{m}\left(P\right)=T_{0}{\left(1+\frac{P}{\prod }\right)}^{1/b}$$ where
$T_{0}=307.9\;K$ is related to the melting temperature under atmospheric pressure, and
∏=431 MPa
and
b=5.65
are empirical constants when recalling the original reference of this model relation.

When considering the barocaloric ‘experiment’, related to the subsequent isothermal path of approaching the $T_{O-C}\left(P\right)$ curve, what is notable is the rise in ${\left[dT_{O-C}\left(P\right)/dP\right]}^{-1}$, which has led to a similar change in the related contribution to entropy
${\Delta s}_{2}$ in Equation (2).

## 5. Conclusions

This report presents the first results on broadband dielectric spectroscopy insights into ODIC-forming neopentyl glycol (NPG) under compression up to the GPa domain. Particular attention was paid to the strongly discontinuous phase transition ODIC–solid crystal. The insights cover static, dynamic, and energy-related properties, namely evolutions of the dielectric constant, DC electric conductivity, and dissipation. Worth stressing are the unique findings related to the pressure-related Mossotti catastrophe-type behavior of the dielectric constant, the novel approach to non-Barus dynamics, and the discussion on fundamentals of dissipation factor changes in NPG. The report significantly addresses the low-frequency domain in BDS-related spectra problems.

A complex phase sequence of compression has been evidenced in the solid crystal phase. Also, a notable pretransitional effect, with critical-like characterization, was found for the solid phase near the discontinuous transition ODIC–crystal. This allows us to ask whether a parallel grain model, initially developed for the premelting effect near the liquid–solid crystal discontinuous transition, is possible for a given phase transition. This model assumes the appearance of solid-state crystalline grains surrounded by quasi-liquid nanolayers. Recently, it has been shown, based on BDS studies, that the critical characteristics of the temperature and pressure changes in the latter are as follows: For the system studied in this work, the parallel system (crystalline grains + liquid nano-layers) could be a pretransitional system composed of crystalline grains surrounded by ODIC nano-layers, with the orientational freedom of molecules and moments coupled with dipoles. The enormous and frequency-dependent difference between the conductivities and dissipation factor in the crystalline and ODIC phases shown in this work may indicate the importance of the mentioned orientational freedom for the translational motions of, for example, percolating ions. It is worth noting here the proximity of the quasi-critical singular pressure in the solid state and the pressure of the discontinuous phase transition, which is also characteristic of the ‘critical nano-layers grain model’ discussed in refs. [31,32,38,40,41]. Finally, summarizing the results of this work, the authors point out their complementarity with the unavailable research results discussing temperature changes under atmospheric pressure. The results presented in this work and ref. [43] also introduce qualitatively new experimental evidence concerning the properties of the ODIC mesophase and the related discontinuous phase transition to the solid crystalline phase. This work also supplements the missing evidence relevant to the potentially exceptional colossal barocaloric effect applications of NPG. The discovery of Mossotti catastrophe-type behavior may be particularly important, since it links the ODIC phase with the Clausius–Mossotii local field and paraelectric-like pretransitional behavior (this work and refs. [43,66]). Hence, obtaining a very high dielectric constant in ODIC-forming material at the discontinuous phase transition can be possible with the appropriate material engineering. This can significantly contribute to the entropy change at the phase transition (see Equation (2)) and, consequently, a significant colossal barocaloric effect.

## Figures and Tables

**Figure 1 materials-18-00635-f001:**
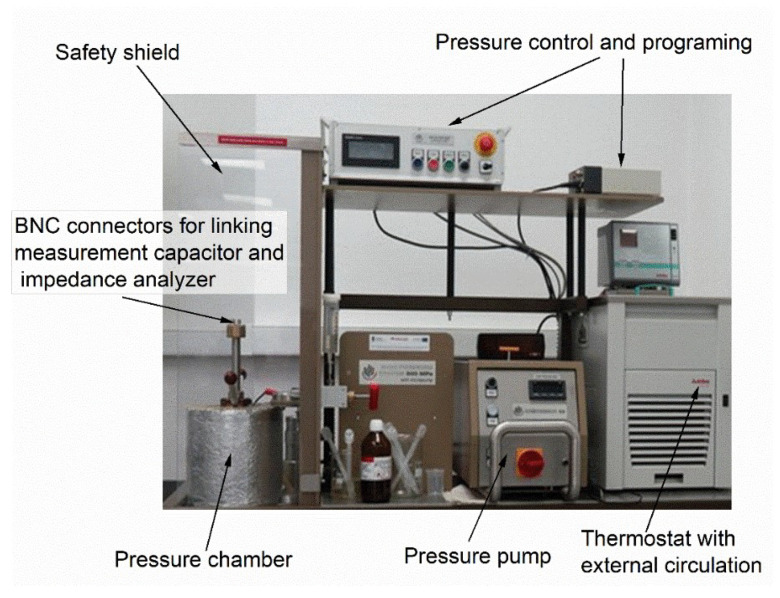
The photo of the pressure setup used for reported experimental studies.

**Figure 2 materials-18-00635-f002:**
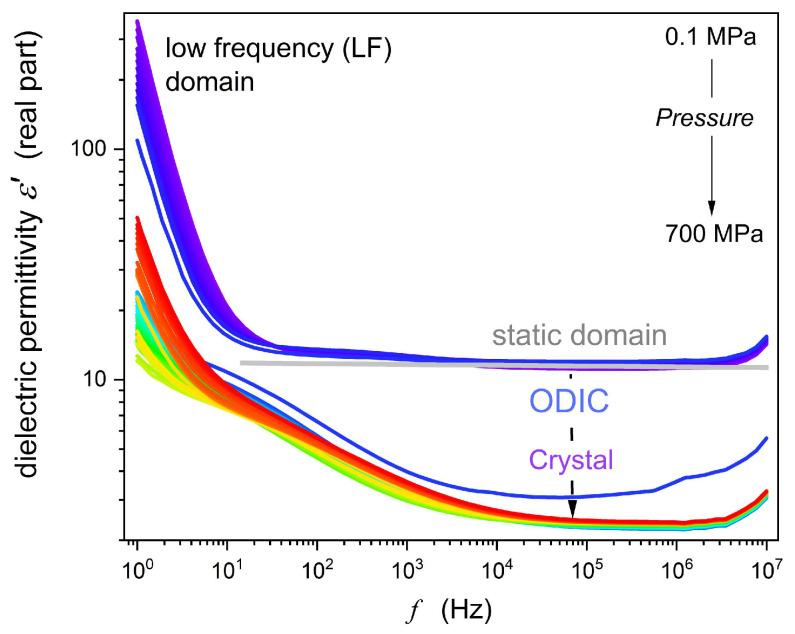
Example of the real part of the dielectric permittivity spectrum obtained in high-pressure studies in NPG, for T=50 °C.

**Figure 3 materials-18-00635-f003:**
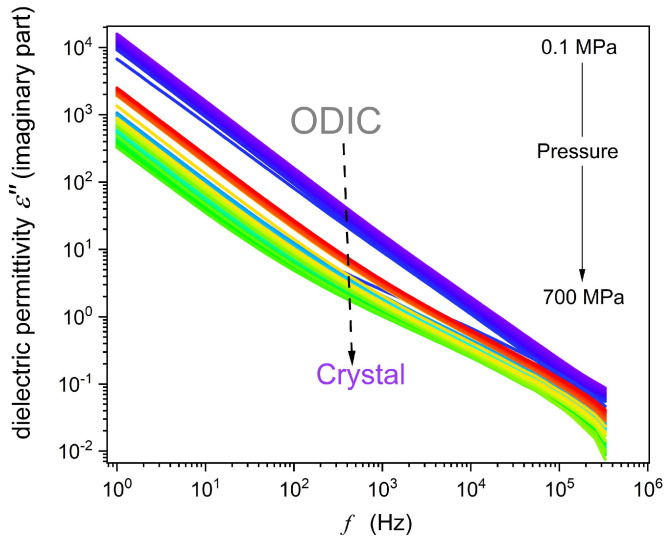
Example of the imaginary part of the dielectric permittivity spectrum obtained in high-pressure studies in NPG, for T=50 °C.

**Figure 4 materials-18-00635-f004:**
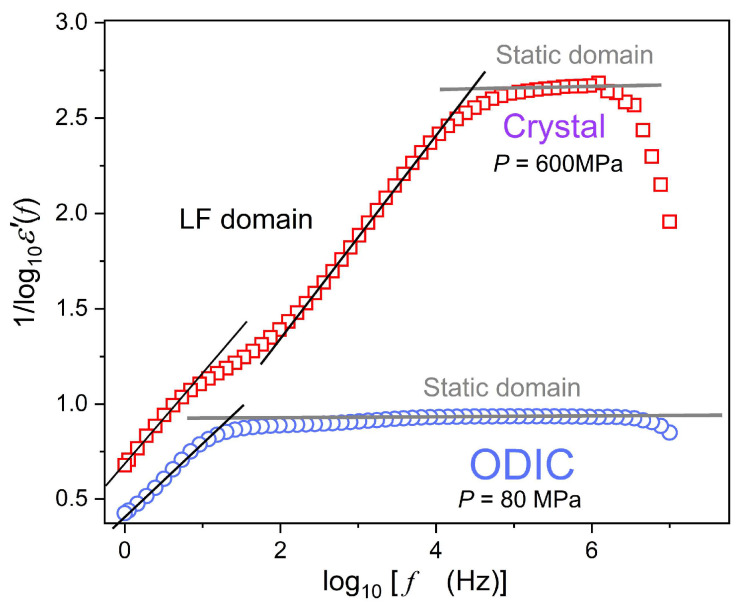
Frequency changes of the real part of dielectric permittivity reciprocal for the selected pressures in the ODIC and crystal phases, based on results shown in Figure 2. Lines in black are plotted to indicate the dominant pattern of frequency changes.

**Figure 5 materials-18-00635-f005:**
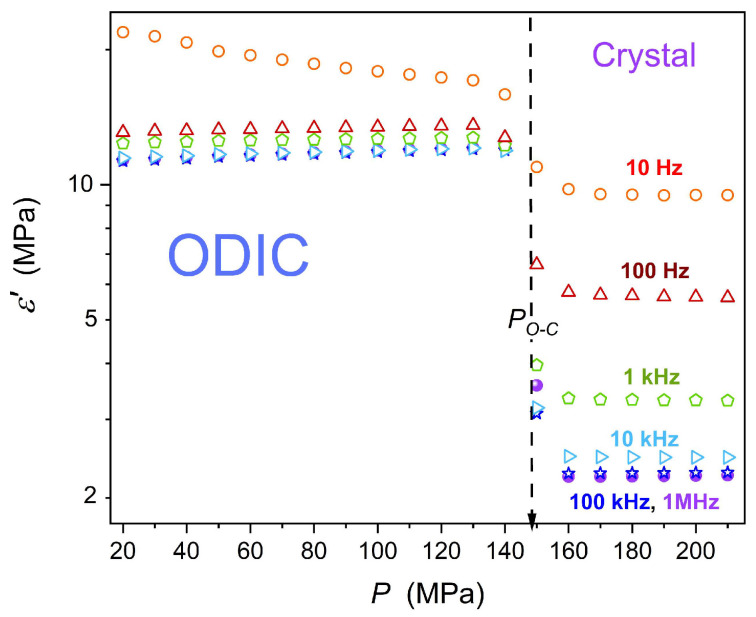
Isothermal (T=50 °C) pressure dependence of the dielectric constant in NPG, for the surrounding of the ODIC—solid crystal transition.

**Figure 6 materials-18-00635-f006:**
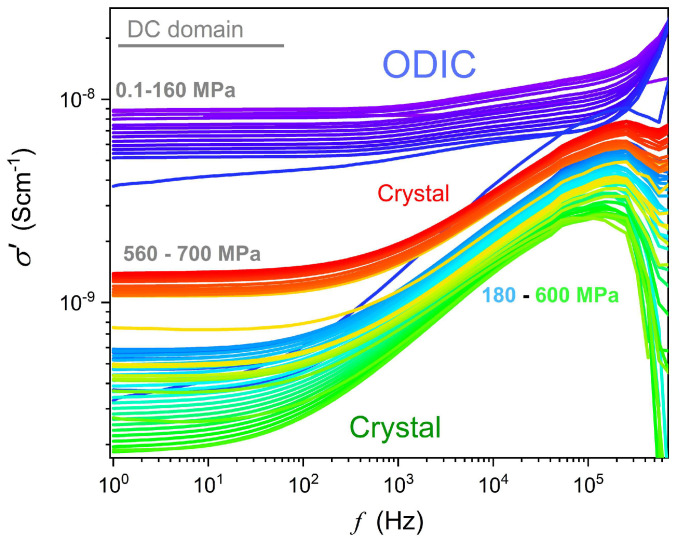
The real part of the electric conductivity spectrum. The DC electric conductivity is related to the horizontal behavior, explicitly appearing in the ODIC phase for f<100 Hz, and in the solid crystal phase, it can be approximated for f=1 Hz. Note: the DC electric conductivity appears as the horizontal domain in such plots.

**Figure 7 materials-18-00635-f007:**
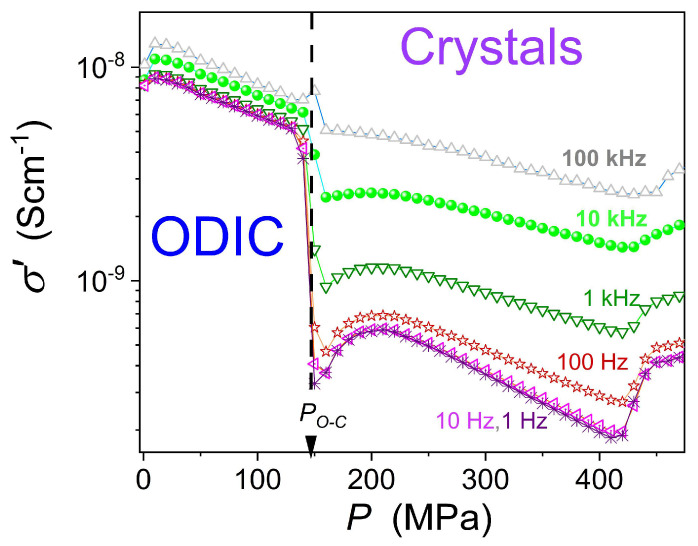
Pressure changes in electric conductivity in the ODIC and solid crystal phases of NPG, for a set of frequencies. The dashed arrow indicates the discontinuous phase transition ODIC–crystal. The solid curve portraying data for the DC conductivity limit is related to Equation (10).

**Figure 8 materials-18-00635-f008:**
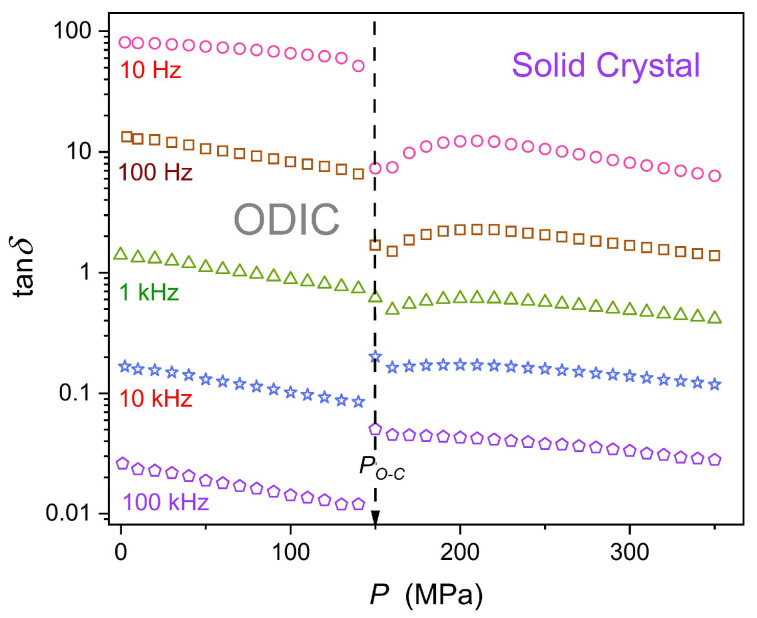
The pressure evolution of the dissipation fact (Equation (12)), based on data presented in Figure 2 and Figure 3, for a set of frequencies in the plot. The semi-log scale enables an insight that is non-biased by decadal changes in values.

**Figure 9 materials-18-00635-f009:**
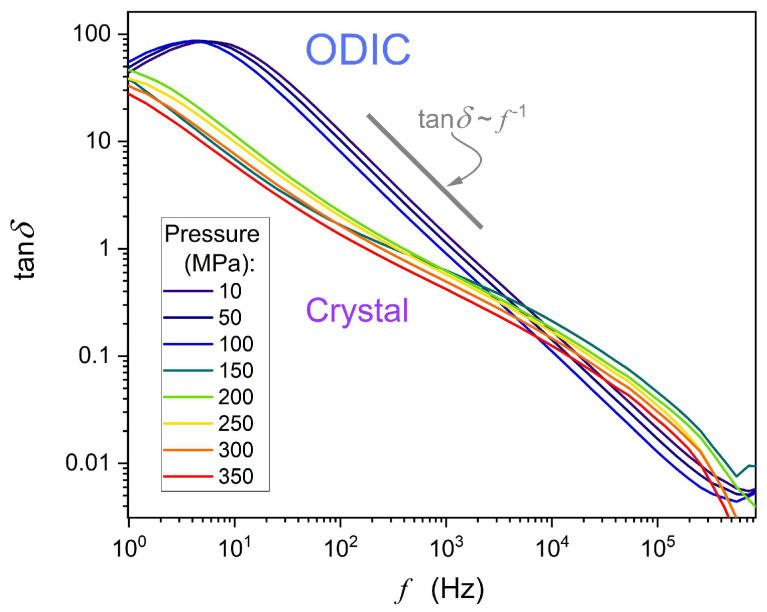
Frequency changes of the dissipation factor
$D\left(f\right)=tan\delta \left(f\right)=\varepsilon^{\prime \prime }/\varepsilon^{\prime }$ in neopentyl glycol (NPG), for selected pressures.

**Figure 10 materials-18-00635-f010:**
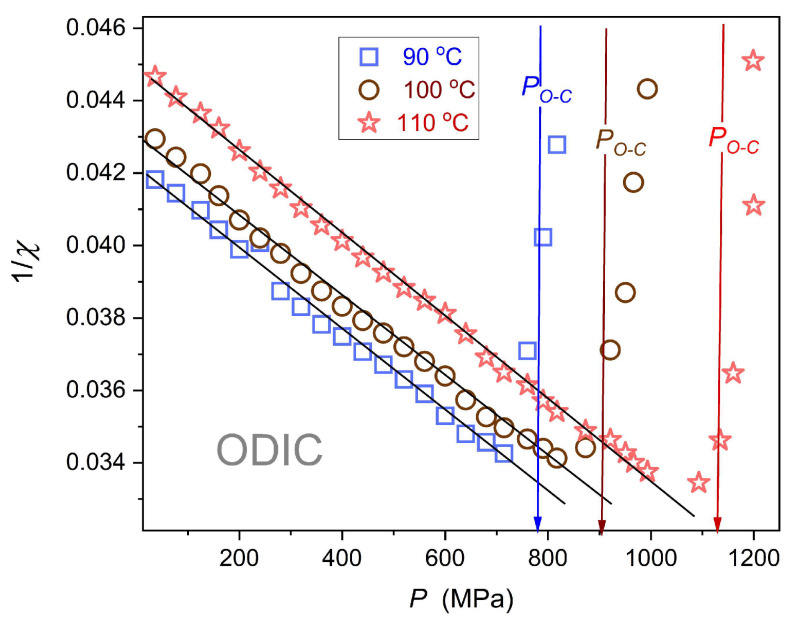
Pressure-related evolution of the reciprocal of dielectric susceptibility
χ=ε−1, for the isotherm T=110 °C, where the ODIC phase is extended up to P~1.2 GPa; the line following the data supports parameterization via the Mossotti catastrophe Equation (13).

**Figure 11 materials-18-00635-f011:**
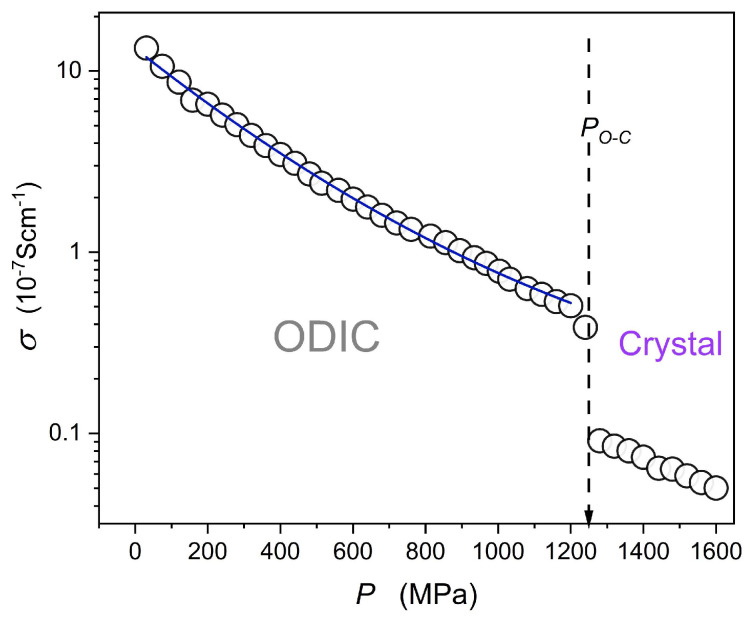
Pressure evolution of DC electric conductivity for the isotherm
T=110 °C, where the ODIC phase is extended up to
P~1.2 GPa.

**Figure 12 materials-18-00635-f012:**
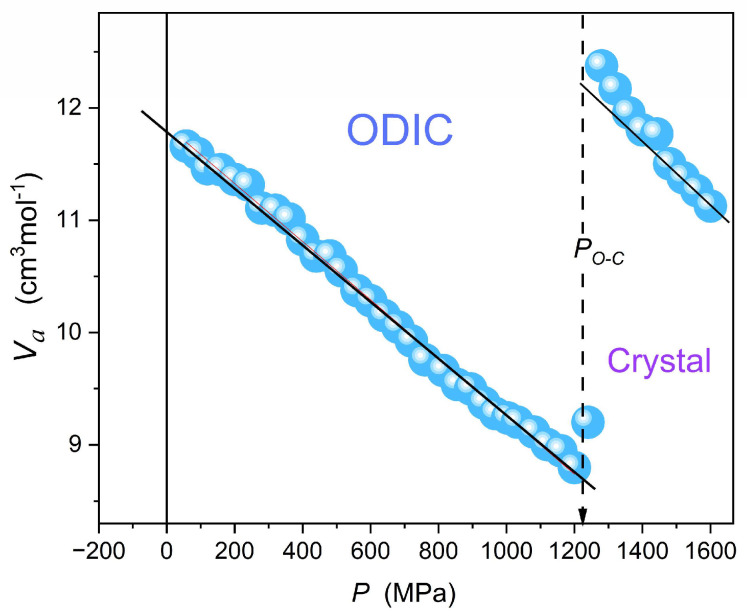
Apparent activation volume pressure changes for the isotherm T=110 °C. The solid line in the ODIC phase is portrayed as $V_{a}\left(P\right)=a+bP=-0.026+11.84P$, for $P_{sp}=-55\;MPa$.

**Figure 13 materials-18-00635-f013:**
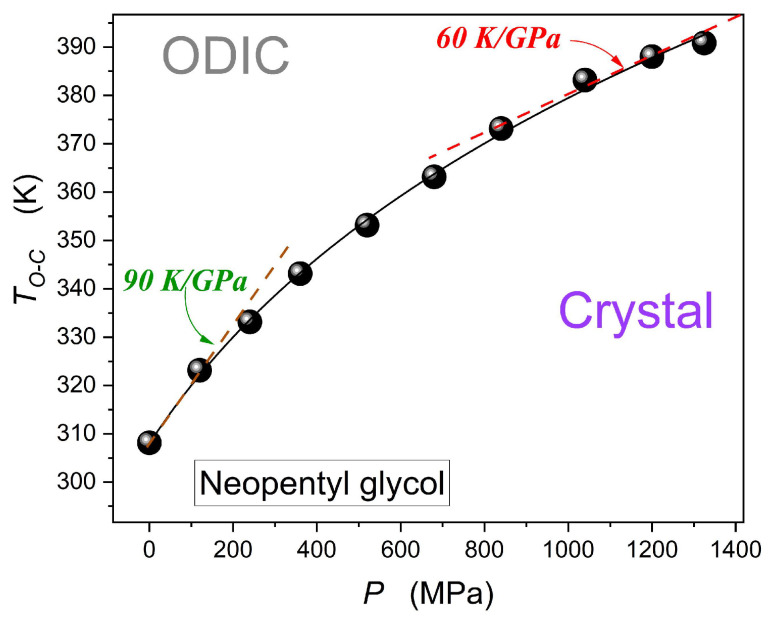
The ODIC–crystal discontinuous phase transition pressure dependence up to GPa domain. The parameterization is related to the Simon–Glatzel Equation (18).

## Data Availability

The original contributions presented in this study are included in the article. Further inquiries can be directed to the corresponding author.

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
