# Peer review of "Preliminary Broadband Dielectric Spectroscopy Insight into Compressed Orientationally Disordered Crystal-Forming Neopentyl Glycol (NPG)"

_materials, 2025, doi:10.3390/ma18030635_

Round 1
Reviewer 1 Report
Comments and Suggestions for Authors
The manuscript offers valuable insights into NPG’s dielectric behaviour and phase transition dynamics, but significant improvements are needed to elevate its scientific rigor and impact. The introduction lacks a cohesive integration of unresolved challenges in the field, which would justify the study’s objectives more convincingly. Experimental methodologies, while detailed, do not sufficiently address sources of uncertainty, alternative approaches, or the broader implications of the chosen techniques. The results sections often present qualitative observations without robust quantitative modelling or thorough mechanistic explanations, limiting the reader’s ability to assess the novelty and significance of the findings. Critical behaviours such as the Mossotti-Catastrophe effect and pretransition dynamics are introduced without rigorous validation through comparative analysis or independent techniques. The discussion of potential applications, while promising, remains superficial and fails to engage deeply with practical limitations or scalability challenges. Finally, the conclusions overgeneralize the study’s contributions, neglecting to contextualize the findings within the broader scientific and industrial landscape, while the references, though relevant, lack diversity in cross-disciplinary insights and comparative studies. Addressing these issues will require a more balanced approach that combines technical depth, quantitative rigor, and broader contextual relevance, ensuring the manuscript meets the highest standards of scientific innovation and clarity.
Below authors can find a detailed section-by section report. I strongly suggest the authors to answer to all the questions raised by the reviewer and insert all the answers properly in the final manuscript.
Abstract
The abstract lacks a coherent narrative, specific results, and broader implications diminishes its ability to fully engage both specialists and general readers. Reorganizing the abstract to balance technical depth with clarity and focusing on its significance for both theory and application would significantly improve its quality. If there is no space in the abstract to properly answer the following questions, insert your answers within the body of the manuscript in the rest of the sections.
A1) How does the study quantitatively advance the understanding of the Mossotti-Catastrophe-type behavior in dielectric materials, and what are the specific experimental uncertainties associated with these findings under GPa-level pressures?
A2) What new mechanistic insights into the colossal bar caloric effect in NPG are provided by this study, and how do they compare with established bar caloric materials in terms of efficiency and scalability?
A3) Can the authors provide explicit comparisons of ODIC-to-solid crystal phase transitions in terms of entropy changes, pressure thresholds, and dielectric behaviour, and how do these compare with theoretical predictions?
A4) What specific role do molecular orientation and dipole dynamics play in the observed changes in the dissipation factor, and how can these be experimentally isolated from external effects like contamination or structural defects?
A5) What are the reproducibility and scalability challenges of this research for industrial applications, particularly in high-pressure regimes beyond 1 GPa, and how does the study address these?
1. Introduction
The section lacks sufficient depth in presenting unresolved scientific challenges and the novelty of the research. A stronger focus on integrating prior findings with current gaps would enhance the section's clarity and relevance, better preparing readers for the subsequent technical content.
1.1) What specific structural features of NPG’s ODIC mesophase distinguish it from other plastic crystals, and how do these features quantitatively influence its dielectric and bar caloric properties?
1,2) How does the theoretical framework for ODIC phase behaviour integrate recent advancements in critical phenomena physics, and what unique challenges does NPG present in this context?
1.3) How do the entropy changes associated with NPG's phase transitions quantitatively compare to other CBE materials, and what molecular mechanisms drive its exceptional performance?
1.4) What are the thermodynamic and kinetic constraints that govern the ODIC-to-solid crystal transition under high pressure, and how are they experimentally validated?
1.5) How do molecular level orientational and translational dynamics in NPG’s ODIC phase contribute to its dielectric anisotropy, and what experimental methods can resolve these dynamics with higher precision?
1.6) Some recent technology developments are missing and should be taken into consideration in this study such as: metastructures [Inverse-designed metastructures that solve equations, Science 363 (6433), 1333-1338, 2019] and nanoparticles [Targeted dielectric coating of silver nanoparticles with silica to manipulate optical properties for metasurface applications, Materials Chemistry and Physics, 126250, 2022].
2. Temperature-related BDS studies under atmospheric pressure in neopentyl 89 glycol
This section impact is diluted by a lack of detailed experimental data integration and limited exploration of molecular mechanisms. A deeper connection between theory, experimental observations, and molecular dynamics would elevate the section's scientific rigor and clarity, offering a more comprehensive view of NPG's unique properties.
2.1) How does the relaxation time behaviour of NPG compare quantitatively to other ODIC materials under similar temperature conditions, and what unique molecular interactions explain these differences?
2.2) To what extent do the deviations from Arrhenius or VFT behaviour reveal new insights into the coupling between translational and rotational dynamics in the ODIC phase?
2.3) How are low-frequency dielectric responses, particularly ionic contributions, experimentally disentangled from intrinsic molecular dynamics in NPG, and what are the sources of error?
2.4) What are the precise roles of dipole-dipole interactions and intrinsic defects in shaping the dielectric loss spectrum of NPG, and how can they be further resolved experimentally?
2.5) How does the interplay between proton hopping mechanisms and molecular orientation freedom affect the DC conductivity in the ODIC and crystalline phases, and how do these mechanisms evolve near critical temperatures?
3. Materials and Methods
The section would benefit from greater emphasis on the reasoning behind experimental choices, a more detailed discussion of challenges and uncertainties, and enhanced integration of the provided figures into the narrative. Addressing these gaps would significantly strengthen the section and ensure that the methodologies are transparent, reproducible, and scientifically robust.
3.1) How are the effects of parasitic capacitance quantitatively minimized during high-pressure BDS measurements, and what systematic errors remain unaddressed?
3.2) What are the reproducibility and reliability statistics for the experimental setups at pressures beyond 1 GPa, and how do these statistics inform the confidence intervals of the measured dielectric properties?
3.3) How does the use of Teflon tubes and Plexol as pressure-transmitting media affect the dielectric behaviour of NPG, and what comparative experiments were conducted to rule out media-related artifacts?
3.4) What are the specific challenges associated with maintaining isothermal conditions during high-pressure measurements, and how were these addressed experimentally?
3.5) How do the dimensional tolerances of the measurement capacitor and sample positioning impact the uniformity of the electric field, and what sensitivity analyses were performed to account for these factors?
4. Results and Discussion
The Results and Discussion section can be significantly enhanced by integrating advanced quantitative models that provide explicit comparisons between experimental data and theoretical predictions, as well as by deepening the mechanistic analysis through molecular dynamics simulations or high-resolution spectroscopic techniques to uncover the underlying physical processes. Benchmarking NPG's properties against other ODIC materials or bar caloric compounds would emphasize its unique contributions, while a critical assessment of the assumptions and limitations of empirical models, with refinements or alternatives, would improve the robustness of the findings. Finally, contextualizing the results within broader scientific and technological frameworks and proposing specific avenues for future research would strengthen the relevance and forward-looking impact of the study. See below a more detailed review for each sub-paragraph.
4.1 Real part of dielectric permittivity, moderate pressures
The sub-paragraph scientific impact is weakened by a lack of quantitative rigor and insufficient exploration of underlying mechanisms. The addition of molecular-level insights, explicit numerical trends, and comparisons to analogous systems would enhance the robustness of this analysis.
4.1.1) How does the molecular orientation in NPG’s ODIC phase influence the static domain of ε', and can it be quantified through modelling or complementary experimental techniques?
4.1.2) What is the role of pressure-induced molecular rearrangements in shaping the low-frequency dielectric boost, and how do these compare to theoretical predictions?
4.1.3) How does the observed pretransition rise in ε' in the solid phase relate to known critical phenomena in phase transitions, and what are the potential scaling laws?
4.1.4) What corrections are applied to ε' data to account for extrinsic effects such as electrode polarization, and how do these affect the interpretation of LF domain behaviour?
4.1.5) How does the dielectric response in the static domain vary with pressure increments, and are there any indications of nonlinearities or hysteresis effects?
4.2. Electric conductivity, moderate pressures
This sub-paragraph suffers from a lack of quantitative depth and insufficient exploration of the molecular and dynamic mechanisms driving the observed trends. A more rigorous integration of experimental data with theoretical models and comparative analyses would significantly strengthen the scientific quality of this section.
4.2.1) How does the molecular structure of NPG contribute to its observed DC conductivity trends under moderate pressures, and how does this compare to other ODIC-forming materials?
4.2.2) What are the precise activation volumes for ionic and molecular transport in the ODIC phase, and how do these vary as pressure approaches the ODIC-crystal transition point?
4.2.3) To what extent does pressure alter the distribution and dynamics of defects or free carriers in NPG, and how are these effects reflected in the conductivity spectrum?
4.2.4) How does the interplay between frequency-dependent ionic motion and molecular reorientation affect the shape of the conductivity spectrum in the ODIC and solid crystal phases?
4.2.5) What are the critical exponents or scaling laws governing the pretransition rise in conductivity near the ODIC-crystal phase transition, and how do they align with theoretical models?
4.3 Dielectric loss factor changes
This sub-paragraph lacks sufficient depth in its mechanistic explanations and quantitative modelling. Addressing these gaps with theoretical scaling laws, molecular insights, and comparative analyses would significantly enhance the rigor and impact of this section.
4.3.1) What molecular-level processes dominate the exponential pressure dependence of the dielectric loss factor in the ODIC phase, and how can they be experimentally isolated?
4.3.2) How does the frequency-dependent behaviour of D correlate with specific ionic and dipolar mechanisms, and what are the critical thresholds for these transitions?
4.3.3) What are the theoretical underpinnings of the pretransition rise in D near the ODIC-crystal phase boundary, and how does this behaviour compare with other dielectric systems under compression?
4.3.4) How does the dissipation factor in NPG evolve under combined temperature and pressure variations, and what are the implications for its potential applications?
4.3.5) How does the observed frequency-dependent dissipation factor relate to the broader dielectric relaxation spectrum, and are there identifiable critical frequencies for molecular or ionic processes?
4.4 Dielectric constant: extreme pressures tests
This section lacks quantitative rigor and detailed mechanistic explanations. A more thorough integration of theoretical models, comparative analysis, and experimental validations would significantly elevate the scientific robustness and broader relevance of this sub-paragraph.
4.4.1) What molecular-level interactions drive the nonlinear evolution of the dielectric constant at extreme pressures, and how do they deviate from moderate-pressure behaviour?
4.4.2) How robust is the Mossotti-Catastrophe model in describing the susceptibility divergence in NPG, and what alternative models could better capture the observed trends?
4.4.3) How does the extrapolated critical pressure (P⁺) compare to experimentally observed transitions or anomalies, and what are the implications for the material's stability?
4.4.4) What roles do compressibility changes and structural anisotropy play in the divergence of dielectric susceptibility at extreme pressures?
4.4.5) How do the dielectric properties at extreme pressures correlate with potential structural transformations or defects in NPG, and can these be probed through complementary techniques such as X-ray diffraction?
4.5 DC electric conductivity: extreme pressures test
This section is undermined by a lack of quantitative modelling, detailed mechanistic explanations, and validation of proposed trends. A more rigorous exploration of molecular mechanisms and comparative analyses with other systems is necessary to elevate the scientific significance of these findings.
4.5.1) What molecular-level processes contribute to the observed deviations from linear Barus behaviour, and how do these processes vary across the pressure spectrum?
4.5.2) How does the pressure-dependent activation volume in the super-Barus regime correlate with structural changes or defect dynamics in NPG?
4.5.3) What are the specific limitations of the Barus equation in describing DC conductivity trends at extreme pressures, and how can these limitations be addressed with alternative models?
4.5.4) How do anisotropic molecular interactions in the ODIC phase influence ionic and electronic transport properties under extreme pressure conditions?
4.5.5) What complementary experimental techniques (e.g., impedance spectroscopy, X-ray analysis) can be used to validate the proposed pressure-dependent conductivity mechanisms in NPG?
4.6 Pressure dependence of the ODIC-Crystal discontinuous phase transition temperature
The sub-paragraph mechanistic insights are limited, and the reliance on a single model without experimental cross-validation weakens its scientific depth. Expanding the analysis to include molecular dynamics, alternative models, and broader contextual comparisons would substantially enhance the robustness and impact of this section.
4.6.1) What molecular or lattice rearrangements dominate the ODIC-crystal phase transition, and how do these change as pressure increases?
4.6.2) How does the Simon-Glatzel parameterization compare to other models in predicting the pressure dependence of the transition temperature, and what are its limitations?
4.6.3) What is the role of compressibility and molecular anisotropy in shaping the transition temperature trends, and how are these experimentally quantified?
4.6.4) How do entropy changes across the ODIC-crystal transition in NPG compare with those of other bar caloric materials, and what implications does this have for its thermal efficiency?
4.6.5) Are there any indications of metastable phases or phase coexistence near the transition pressure, and how might these impact the observed thermodynamic behaviour?
4. Conclusions
The Conclusions section lacks sufficient depth in addressing the broader implications, limitations, and future directions. Its overgeneralization and limited engagement with practical and theoretical challenges dilute its scientific impact. A more rigorous and balanced approach would strengthen the overall quality and relevance of this section.
4.1) What are the specific limitations of this study in accurately characterizing the Mossotti-Catastrophe behaviour, and how can these be addressed in future experiments?
4.2) How does the critical-like pretransition behaviour in NPG compare quantitatively to similar phenomena in other ODIC systems, and what does this imply about the universality of these effects?
4.3) What are the thermodynamic and structural trade-offs in utilizing NPG for energy-efficient refrigeration systems compared to other bar caloric materials?
4.4) How can broadband dielectric spectroscopy be further optimized to probe higher frequency domains or more extreme pressure regimes in NPG?
4.5) What specific advancements in material design or experimental methods are required to transition the findings of this study into scalable industrial applications?
Author Response
First, we are grateful to the Reviewer for his great work and efforts, visible in the extensive size of the opinion and in the deep analysis of each part of the report. The size of the review and the insightful comments indicate that the research and results presented have been of particular interest. We are impressed by this exceptional appreciation.
However, we also have a certain problem with responding to some of the issues indicated in the review. The first issue is the preambles with which each part of the review begins, referring to the subsequent sections of the publication. These preambles are formulated in an unusually categorical and decisive way. The suggestions contained in this way are incomprehensible to us, especially since their justification is not presented in the detailed comments that follow. As specialists who have been dealing with high pressure, dielectric spectroscopy and soft matter for decades, we do not feel comfortable with this approach. We understand that it may result from interest in the work and the topic, but we feel a certain discomfort here.
We would also like to point out that this work is:
- the first record of pressure dielectric tests, additionally covering dynamic and static properties, not only in NPG but in any plastic crystal material, including ODIC-forming ones’
- this research enters the GPa area, which poses huge challenges because pressure tests in this area for soft matter systems are a special challenge that only a few laboratories can cope with
Our goal was to show such special results that support the data analysis methods implemented in the last decade in our works, allowing for a deeper interpretation of the results. We emphasize this issue, because there are many suggestions in the review to perform additional comprehensive supplementary studies as well as simulations and modeling (goal and scope to be defined) in this work. The latter issues should be especially emphasized because it means working significantly beyond the current state of knowledge for ODIC-forming materials and also discontinuous phase transitions, where the cognitive situation still refers in essence to the over hundred-year-old works of Clapeyron and Clausius and Lindemann.
For us, these comments are rather a great research program that solves many cognitive puzzles and problems, which is not possible within the scope of the research data and publications. Implementing comments in a review essentially means overcoming previously unsurmountable cognitive barriers. Such a program would be possible to implement through the cooperation of our team and the Reviewer #1 team after several project studies. We can take up this challenge .
We write about these issues to explain why we respond to many of the detailed review comments in the way we do. To emphasize the point of the publication, which is only about issues (i), (ii), (iii), we have added the word ‘Preliminary’ to the title.
Reviewer #1
- Comment Reviewer #1: for the Abstract the following general comment is formulated: ‘The abstract lacks a coherent narrative, specific results, and broader implications diminishes its ability to fully engage both specialists and general readers. Reorganizing the abstract to balance technical depth with clarity and focusing on its significance for both theory and application would significantly improve its quality. If there is no space in the abstract to properly answer the following questions, insert your answers within the body of the manuscript in the rest of the sections.
Response: These general comments, are followed by 5 explicit points, suggesting explanation of issues beyond the target of the report.the report is devoted to the first ever BDS studies in NPG, which are maybe the send ever such result in any ODIC forming materials. only this and nothing more. The Abstarct simply presents basic facts, without speculative comments.
- (Introduction) Comment Reviewer #1: ‘The section lacks sufficient depth in presenting unresolved scientific challenges and the novelty of the research. A stronger focus on integrating prior findings with current gaps would enhance the section's clarity and relevance,.. ‘ It is further followed by 6 points, explicitly addressing ‘lacking issues’
Response: This ‘general criticism’ is commented/explained above. Please note the scope of the presented research given in the title of the report.
- (Remarks on NPG temperature BDS tests) Comment Reviewer #1: ‘This section impact is diluted by a lack of detailed experimental data integration and limited exploration of molecular mechanisms.,,,, ‘, followed by some specific issues specified in 5 points
Response: In our opinion, in view of the purpose of the work and the presented results, this statement is unjustified and inappropriate.
3a) Comment Rev. #1: ‘How does the relaxation time behaviour of NPG compare quantitatively to other ODIC materials under similar temperature conditions, and what unique molecular interactions explain these differences?’
Response to 3.1: in our opinion this is a tautological question, because in the given section and in the Introduction the nature of the ODIC state, namely free rotation of molecules/dipole translational freezing in a crystalline network is indicate – and answers the given question. Repeating this argument will not improve the clarity of the report.
- . Comment Rev. #1: ‘To what extent do the deviations from Arrhenius or VFT behaviour reveal new insights into the coupling between translational and rotational dynamics in the ODIC phase?
Response to 3.2: Reviewer #2 mixes two separate issues here. The first is the inadequacy of the description of the changes in the CSU of the rectification or DC electrical conductivity using the Arrhenius and VFT relations. This issue is raised in the given section and is based on clear references, e.g. in Physical Review (works by S.J. Rzoska, J.Ll. Tamarit et al.) and then in-depth discussion in the works of A. Drozd-Rzoska et al. in Scientific Reports and Progress in Materials Science. These references are explicitly given and repeating these results is inappropriate for the consistency of the given work. Translational – orientational coupling – or in complex soft systems decoupling – is another problem related to the so-called fractional Debte-Stokes-Einstein law.
- Comment Rev. #1: ‘How are low-frequency dielectric responses, particularly ionic contributions, experimentally disentangled from intrinsic molecular dynamics in NPG, and what are the sources of error?
Response to 3.3: this is again a general question, which has not been answered yet explicitly in any report. The part about the error is unclear, because in the Experimental part it is clearly written that the data were recorded with a resolution of 5-6 digits, with high stability of conditions (P,T). The authors express the following comment in the Results and Discussion section, (starting from 279 line):
‘ The strong rise of dielectric permittivity is essentially linked to ‘generally recalled ‘ionic species’ or ‘contaminations [68-75]. Nevertheless, they can also explaine by translational shift, in respect to the average position, for basic molecules of a given system/material [50].Worth recalling, is the still existing challenge for dielectric permimittivity parameterization in the LF domain, both for and [66-73]. Notable that the latter is directly related to electric conductivity, as indicated above’
2.4 Comment Rev. #1: ‘What are the precise roles of dipole-dipole interactions and intrinsic defects in shaping the dielectric loss spectrum of NPG, and how can they be further resolved experimentally?’
’ Response: Once more, the question appears, which is essential, but nobody explicitly and convincingly answered it in numerous dielectric studies in ODIC forming systems. But this report and published in Sept. 2024 report on temperature studies allows for a bit surprising answer to this question - it is now explicitly given in Conclusions. Namely, the explicitly evidence Mossotti – Catastrophe behavior, which is directly related to Clausius-Mossotti local field suggest that dipole-dipole interactions are weak/very weak – because it is the paradigm of the Clausius – Mossotti local field.
- Comment Rev. #1 ‘How does the interplay between proton hopping mechanisms and molecular orientation freedom affect the DC conductivity in the ODIC and crystalline phases, and how do these mechanisms evolve near critical temperatures?
Response to 2.5 The question arises if the ‘proton hoping’ mechanism exist. In our opinion it is a speculative hypothesis introduced ad hoc solely on limited dielectric (BDS) data by Wuebbbenhorst et al. (see ref. [59]). We do not know convincing experimental arguments there. Following this we do not address or use this concept in the given report, because it is not significant for the results discussion.
- Comments of Reviewer #1 on Methods and Materials section
General remarks of the Reviewer #1: ‘ The section would benefit from greater emphasis on the reasoning behind experimental choices, a more detailed discussion of challenges and uncertainties, and enhanced integration of the provided figures into the narrative. ‘
General Response: this remark is not clear to the authors, because as answered below to specific questions extensive, professional comments are given in this section, namely:
3.1 Reviewer #1: ‘How are the effects of parasitic capacitance quantitatively minimized during high-pressure BDS measurements, and what systematic errors remain unaddressed?
Response: the following explanation has been added in the text
Lines 238 – 241 ‘
‘The parasitic capacitances were removed following the protocol of the impedance analyzer producer. Additionally tests using menthol and nitrobenzene as standard liquids with well-3known dielectric parameters were used for the validation.’
It is important that we operate using the complete Novocontrol set-up with all supporting facilities and software support – but we are not sure if this is proper to mention in the report.
3.2 Reviewer #1: What are the reproducibility and reliability statistics for the experimental setups at pressures beyond 1 GPa, and how do these statistics inform the confidence intervals of the measured dielectric properties?
Response: see new comments in lines 242-251: ‘The reproducibility of results in BDS measure in the under-GPa (Method 1) and above-GPa domain depends on the precision of pressure estimation for both methods, as noted above. Nevertheless, the relative error does not differ strongly, when taking into account rising pressure values. The analyzer enables the cumulative analysis of the registered response. Additionally, the results presented are based on three experimental sets; no significant differences in values were detected. For the temperature stabililization, it is always very high for properly designed set-ups. Temperature was scanned within and along the chamber to avoid gradients. Compressing was done in small pressure steps, and a 10-minute pause took place before subsequent measurements. Hence, it cannot change the temperature of the tested sample.
3.3 ‘Reviewer #1: ‘How does the use of Teflon tubes and Plexol as pressure-transmitting media affect the dielectric behaviour of NPG, and what comparative experiments were conducted to rule out media-related artifacts?
Response: See the added comment in lines 216 – 220: ‘The thin tube causes immediate transmission of pressure to the sample from pressurized transmission liquids, as well as avoiding contamination. The usage of Plexol, a liquid specially ‘designed’ for high pressure, guaranteed the same and not high viscosity during processing, which allows the lack of its impact on the measurement process and a destruction of the (expensive) pressure pump when approaching GPa domain,
3.4 ‘’Reviewer #1: ‘What are the specific challenges associated with maintaining isothermal conditions during high-pressure measurements, and how were these addressed experimentally?’
Response: this issue is commented in the given section in a way clear for experimentalists. Note : (i) 25 liter Julabo thermostat with the external circulation (ii) the pressure chamber is a few tens of kg mass of steel , surrounded by a ‘jacket were the liquid circulate, (iii) plus external thermal isolation. Add to this please temperature scans. All these is and was stated in the Methods section.
3.5 Reviewer #1: ‘How do the dimensional tolerances of the measurement capacitor and sample positioning impact the uniformity of the electric field, and what sensitivity analyses were performed to account for these factors?
Response: Note that the macro-gap between capacitor plates and the distance, the fact that they are made from Invar, and the distance between plates is kept via precise quartz elements beyond the measurement domain. All these is explicitly stated in the Methods section.
Please Note: that (very) often in dielectric/BDS measurements, ‘distant’ elements just within the measurement area and usually d=10 – 50 micrometer – which, in our opinion, can introduce ‘dangerous’ artifacts. We avoid such parasitic impacts
Section Results & Discussion….
4 . Reviewer #1 (General comment) : ‘The Results and Discussion section can be significantly enhanced by integrating advanced quantitative models that provide explicit comparisons between experimental data and theoretical predictions, as well as by deepening the mechanistic analysis through molecular dynamics simulations or high-resolution spectroscopic techniques to uncover the underlying physical processes….’
Response the Reviewer ask for simulations, advanced modeling, relation to other materials – with the stress to barocalorics , and further integrating application using results presented.
In our opinion it is rather an extensive and advanced research program that can lead to (at least) a few high-rank reports that the general comment to this report which targe is (ONLY) the first preliminary report on BDS studies in NPG under compressing.
Note also that in the past there was only one pressure related report in ODIC – forming material, but focused solely on relaxation processes (the authors of the given report + J.Ll Tamarit team).
Below responses to specific suggestions:
4.1 Reviewer #1 ‘ Real part of dielectric permittivity, moderate pressuresThe sub-paragraph scientific impact is weakened by a lack of quantitative rigor and insufficient exploration of underlying mechanisms. The addition of molecular-level insights, explicit numerical trends, and comparisons to analogous systems would enhance the robustness of this analysis.’
Response: for the authors this comment is incomprehensible, also because it is supported by generalized arguments:
4.1.1 Reviewer #1 ‘How does the molecular orientation in NPG’s ODIC phase influence the static domain of ε', and can it be quantified through modelling or complementary experimental techniques?’
Response: this is a general question in the analysis of any dielectric system. However, in the given report the answer seems obvious from the finding of Mossotti Catastrophe behavior . In temperature tests and in the given report in pressure test. It directly indicates to the simples local field Clausius – Mossotti model, where the ‘modeling’ is well and generally knows. For supplementary techniques – this issue is beyond the given report, but we can imagine some spectroscopic methods implemented to samples under different intensities of the electric. It would quite different research than in the given report – requiring in-depth preparation , So we cannot discuss and comment it in the report. It is beyond the topic of the report.
4.1.2 Reviewer #1 ‘What is the role of pressure-induced molecular rearrangements in shaping the low-frequency dielectric boost, and how do these compare to theoretical predictions?’
Response: there is no fundamental difference between the spectrum of complex dielectric permittivity, which obviously makes this comment irrelevant. There are therefore no specific models in this aspect, which is a generally known fact. However, one thing is generally known - compression significantly expands the static domain, which is also generally known and visible in Figure 2. This issue is not the subject of the publication. This issue - as one of the few - was dealt with by the authors of this paper 2-3 decades ago. Referring to these works would immediately lead to exceeding the 15% limit of 'own publications', which is prohibited.
4.1.3 Reviewer #1 ‘How does the observed pretransition rise in ε' in the solid phase relate to known critical phenomena in phase transitions, and what are the potential scaling laws?
Response: We guess that the comment is related to Figure 2, below the relatively ‘short’ static domain, We do not have a concept regarding origin of this phenomenon’. Notable is only the fact that it negligibly influence of the pressure dependence of the real part of dielectric permittivity, only shifting the total value Figure 6). Notwithstanding, this issues is discussed in Figure 4 and comments below. This Figure directly leads to Eq. (9) which surprisingly well portrays frequency changes in the solid Crystal phase. The constitute the contribution of this report to further theoretical modeling. Equations (9) are the scaling laws, which are recalled in this comment.
4.1.4 Reviewer #1 What corrections are applied to ε' data to account for extrinsic effects such as electrode polarization, and how do these affect the interpretation of LF domain behaviour?
Response: as discuss above such effects are shifted by compressing to lower frequencies. They are not visible in Figures and 3, which is clear for any researcher familiar with dielectric physics. So they can be expected well below 1 Hz frequency, beyond the domain and problem discussed in the given report.
4.1.4 Reviewer #1 ‘How does the dielectric response in the static domain vary with pressure increments, and are there any indications of nonlinearities or hysteresis effects?
Response: The hysteresis means changes in behavior on compressing and decompressing, particularly the shift of the discontinuous phase transition pressure or temperature. Generally, pressure studies are (made) always on compressing solely, for technical reasons We noted the problem earlier, and now we are ready to carry out also on decompressing and for 2-3 months such results, with enhanced resolution. We hope it will yield a new standard, not available in other labs. We indicate this problem as ‘future research plans in Conclusions. It is also one of key reasons add the word ‘Preliminary’ in the title.
4.2 Reviewer #1 , General ‘This sub-paragraph suffers from a lack of quantitative depth and insufficient exploration of the molecular and dynamic mechanisms driving the observed trends…’
Response: Please note: They are explicit scaling equations, which takes into account the VFT equation failure. In our opinion they are trends indicator. New and real trends that can serve as the base for future modeling. In fact, the answer to this suggestion is close to the essential progress in the explanation of the complex dynamics of glass forming systems – the grand challenge of the last decades, still waiting for the milestone progress.
4.2.1 Reviewer #1 ‘How does the molecular structure of NPG contribute to its observed DC conductivity trends under moderate pressures, and how does this compare to other ODIC-forming materials?
Response: it is generally known pressure related structural studies, particularly with a reliable temperature control, still constitute a grand challenge. Just in our institute (IHPP PAS) it may be possible if one devotes just to this target a few (or more) months. For ODICs. and any similar plastic crystal materials such data are not available. It is the ‘call for future research’, but to a comment possible to address.
4.2.2 Reviewer #1 ‘ What are the precise activation volumes for ionic and molecular transport in the ODIC phase, and how do these vary as pressure approaches the ODIC-crystal transition point?’
Response: this issue is/was in-depth explained and commented in lines 456 – 496, including the presentation of the activation volume pressure dependence in Figure 12. It is one of the most significant and really novel results of the given report. We're guessing the Reviewer did not notice these Results and Discussion.’
4.2.3 Reviewer #1 ‘To what extent does pressure alter the distribution and dynamics of defects or free carriers in NPG, and how are these effects reflected in the conductivity spectrum?
Response: to answer this comment in-depth studies using other methods is required. Basing solely on dielectric scans, even so advanced as in the given report any addressing this question can be only ad hoc speculative – hence it lacks a meaning. Results presented in the given report can be used as a reference if the mentioned results appear in the future. But this is a grand experimental challenge.
4.2.4 Reviewer #1 ‘How does the interplay between frequency-dependent ionic motion and molecular reorientation affect the shape of the conductivity spectrum in the ODIC and solid crystal phases?
Response: The only way for a reliable response to this question requires pressures related studies in the frequency range reaching GHz domain, so both ‘ionic’ (low frequency) and orientational (related to the primary relaxation loss curve) could e registered. But nowadays it is impossible !!! In pressure studies the frequencies f = 1 – 10 MHZ (depending on the sample characterization) remains a limit !
4.2.5 Reviewer #1 ‘What are the critical exponents or scaling laws governing the pretransition rise in conductivity near the ODIC-crystal phase transition, and how do they align with theoretical models’
Response: This comment can be related only to Figure 7 , where the solid Crystal ‘pretransitional’ effects is shown for the dielectric constant, i.e. the real part of dielectric permittivity for f=10Hz. It is shown by the solid curve portraying data in this plot and by Eq. (10), with the extensive discussion in its surrounding.
4.3 Reviewer #1 General comment: Dielectric loss factor changes : this sub-paragraph lacks sufficient depth in its mechanistic explanations and quantitative modelling….’
Response: Please note – it is the first ever such evidence not only in NPG but also in any plastic crystal material, including temperature studies. The evidence is explicit, and can serve as the reference for further research.
4.3.1 - 4.3.5 Reviewer #1 – please see the response related to these 5 questions below
Response: these 5 question are very important and in-depth address possible important issues. But it is the first evidence of this phenomenon, particularly under pressure. Please, do not expect that at this preliminary stage of research we answer all relevant quetions.
We manage to yield quite extensive comment and further progress needs further research.
4.4 Reviewer #1: Dielectric constant: extreme pressures tests. General comment: ‘ This section lacks quantitative rigor and detailed mechanistic explanations. A more thorough integration of theoretical models,
Response: the above decisive remarks are repeated in this report. In our opinion they are not appropriate.
4.4.1 – 4.4.5 Reviewer # 1 See comment below
Response: The Reviewer does not notice earlier temperature studies at atmospheric pressure in cyclooctanol and NPG, although these references are given. In this work we have a similar phenomenon as a function of pressure. Together this is an unambiguous reference to the Curie-Weiss type of paraelectric changes and to the Clausius-Mossotti local field model. Without taking into account this fact - probably universal for ODIC forming material questions 4.1.1 -4.1.5 are not possible to answer adequately, and from these questions it is clear that the Reviewer does not take into account this proven experimental issue.
Such a picture clearly defines the location of the phenomenon in the Physics of Critical Phenomena, where the microscopic picture on which Reviewer focuses is of secondary importance. It is a fundamental paradigm of this field of physics,
4.5 Reviewer #1 ‘DC electric conductivity: extreme pressures test . General comment: ‘This section is undermined by a lack of quantitative modelling, detailed mechanistic explanations, and validation of proposed trends. A more rigorous exploration of molecular mechanisms and comparative analyses with other systems is necessary to elevate the scientific significance of these findings.’
Reponse: Once more ‘hard’ and in our opinion not approapriate comemntg. It is detached from the fact that we are showing here the first experimental evidence and expecting that we will immediately explain everything, including a deepened model analysis. The answer is not possible without ad hoc speculations, at the given stage of the research.
Reviewer #1 suggestions 4.5.1 4-5.5
Response: a program for future research and perhaps issues that have seemed intriguing, not issues that can be addressed in any credible and non-speculative way in the context of the presented results – preliminary and the only ones of their kind so far. Two issues can be addressed
- the accusation of the lack of scaling relations is inappropriate because they are significant part of the given report. Some of them are beyond the existing state of the art. For example, see relation (16) which is qualitatively new and results from an innovative and very clear discussion of the changes in the activation volume
- Further studies on the interaction between translational and orientational processes require, first of all, BDS studies with pressure in the range of ~ 1–3 GHz. This is not possible at present - currently high pressure experiments are limited to 3 – 10 GHz, depending on the electric characterization of the tested system.
4.6 Reviewer # 1 Pressure dependence of the ODIC-Crystal discontinuous phase transition, General Comment: ’The sub-paragraph mechanistic insights are limited, and the reliance on a single model without experimental cross-validation….//
Response: This statement is related to the insight well beyond the current state of the art and the scope of the given report. The generalization contained in this statement is in our opinion not appropriate.
4.6.1-4.6.2 Reviewer # 1, see the response below
Response: We think that the Reviewer does not know how great and difficult a challenge it is to obtain such a phase transition temperature dependence as in this publication – especially when we reach the GPa domain.
Further, the Reviewer expects an in-depth discussion of the Simon–Glatzel equation and other alternative equations, as well as a reference to such issues completely outside of such a relationship. Such issues may be pointers for further research, but in no way are and cannot be addressed on the basis of the research to which this work is devoted.
4.7 Reviewer # 1, Conclusions. General comemnts: ‘The Conclusions section lacks sufficient depth in addressing the broader implications, limitations, and future directions. ‘
Response: I feel great discomfort once again reading criticism that has no connection with the scope of research and content of this work. In some cases comment raises (as non-existent) issues that are clearly addressed in this report.
4.7.1 Reviewer # 1 ‘What are the specific limitations of this study in accurately characterizing the Mossotti-Catastrophe behaviour, and how can these be addressed in future experiments?
Response: There are no such limitations. Up to now – in addition to the given work, there are two more monocular reports – in ODIC-forming cyclooctanol and NPG, vs. temperature under atmospheric pressure. References to these works were and are clearly given in Results and Discussion and Conclusions. Future experiments: test in other systems.’
4.7.2 Reviewer # 1 How does the critical-like pretransition behaviour in NPG compare quantitatively to similar phenomena in other ODIC systems, and what does this imply about the universality of these effects?
Response: The existing evidence was cited above. The obvious extension is the study of the nonlinear dielectric effect, i.e. changes in the dielectric constant in a strong electric field, and such studies are clearly shown in the already cited studies in ODIC forming cyclooctanol.
4.7.3. Reviewer # 1 ‘What are the thermodynamic and structural trade-offs in utilizing NPG for energy-efficient refrigeration systems compared to other bar caloric materials?
Response: this topic is beyond the scope of the given report (see its tile). But it addressed in numerous reports based on studies on NPG under atmospheric pressure, recalled in the given paper.
4.7.4 Reviewer # 1 ‘How can broadband dielectric spectroscopy be further optimized to probe higher frequency domains or more extreme pressure regimes in NPG?
Response: BDS measurement, for frequencies limited to 100kHz can be extended ca. 4GPa. Such facility is in Russia, so now and probably for many next years it is unavailable. In our lab the current limit is 2.4 GPa – but it is a challenge for focused project research. The next significant regime is the extension of pressure to 1-3 GHz, i..e. well above the current max 10MHz limit. We considered such development in our Institute of High Pressure Physics 0 and it is possible. But it requires 2 -3 years of focused team work. We did not succeed in searching the financial support.
We think, that all these are technical issues, not proper for the given report.
4.7.5 Reviewer # 1 : ‘) What specific advancements in material design or experimental methods are required to transition the findings of this study into scalable industrial applications?
Response: it is a very good question. The discovery of the Mossotti Catastrophe behavior as possibly universal in the ODIC phase can suggest that for the ODIC – Solid Crystal phase transition, a focused material engineering , can yield an enormous value of dielectric constant in the ODIC phase and then the enormously large change of dielectric constant at the transition. Following the Burns model, mentioned in the Introduction, it can significantly contribute to entropy change and then the barocaloric effect.
In conclusion, some comments on this opinion have yielded discomfort for us as specialists in the given field (from decades). But we appreciate the great effort of the Reviewer and the interest in the results presented.
Reviewer 2 Report
Comments and Suggestions for Authors
Review Report:
In this manuscript by Drozd-Rzoska et al. demonstrated ODIC forming of NPG using broadband dielectric spectroscopy. Their study provided a decent evolution overview of dielectric constant, DC conductivity during the phase transition.
My comments are copied below. I use the following abbreviations, P-page number; L-line number.
Specific Comments:
1. P2-L40: Please introduce BCE.
2. P5-L84: Provide further technical details of the pressure pump.
3. P6-L227: Provide a few references related to available low MHz studies. What were the maximum frequencies achieved in those studies?
4. P8-L261: It would be great to provide an error bar for the frequency change and also do the same for Figure 5.
5. P10-L322: Is this ODIC to Crystal transition frequency dependent? Can you provide a zoomed inset?
Author Response
- Comment: ‘P2-L40: Please introduce BCE.
Response: it has been corrected.
- Comment: ‘ P5-L84: Provide further technical details of the pressure pump’.
Response: The pressure pump appeared in P5-L184. Generally, the design is a confidential issue of the producer, and we can only comment this issue as users. Namely see the following comment, starting from the line 184:
‘The pressure systems were technically designed and constructed by UnipressEquipment. The pressure pump system consisted of two complementary pumps. The first one, with the large volume of the liquid transmitting pressure, enabled compressing up to even 900 MPa, with steps. The switch to the ‘micro-pump’, at any established pressure, enables ‘subtle’ pressure changes in the range 200 MPa, with the resolution . ‘
- Comment: ‘ P6-L227: Provide a few references related to available low MHz studies. What were the maximum frequencies achieved in those studies?‘
Response: Note the supplementation in lines 234 -236, with references:
‘For limited cases, it is 10MHz, but most often: 1- 3MHz.The terminal pressure depends on the capacitance-resistivity-frequency chart for the given impedance analyzer with respect to the tested sample parameters. ‘
- Comment: ‘P8-L261: It would be great to provide an error bar for the frequency change and also do the same for Figure 5.
Response: Studies were carried out using the Novocontrol impedance analyzer, the world-best such type of equipment, as we know from our long practice. The frequency resolution error is well below 1 Hz, so it is not significant in any figure and analysis.
The unique features of this analyzer are presented on Novocontrol GmBH page, which is easily available.
- Comment: ‘P10-L322: Is this ODIC to Crystal transition frequency dependent? Can you provide a zoomed inset?
Response: It is explicitly the discontinuous phase transition, so it cannot be ‘frequency’ dependent. See Figures 5, 6, 7, which present results for a set of frequencies – and there is no frequency dependence. It is exactly what the Reviewer asks.
The frequency-dependence appears only deeply in the solid crystal phase, as visible in Figure 7. It is a crystal-crystal transformation. It is new, and we do not call it ‘phase transition in the report,
Round 2
Reviewer 1 Report
Comments and Suggestions for Authors
The responses provided to Reviewer #1’s comments reflect a commendable effort to address the feedback in detail; however, several aspects of the tone and content could be improved to enhance the effectiveness and professionalism of the rebuttal. Below, I outline key areas where improvements could be made, focusing on the tone, the handling of scope-related limitations, and the engagement with the reviewer’s broader scientific concerns.
First, the tone of the responses often appears defensive and dismissive. Phrases like "strange," "inappropriate," or "detached" detract from the professionalism of the rebuttal and may alienate the reviewer. It is essential to maintain a neutral and constructive tone, even when disagreeing with the reviewer’s comments. Acknowledging the reviewer’s efforts and engaging with their feedback respectfully, even when the points raised are considered beyond the scope of the paper, would demonstrate a collaborative and professional attitude.
Second, the repeated emphasis on the preliminary nature of the study is a valid point, but it could be framed more constructively. Instead of dismissing questions or suggestions as "beyond the scope" or "not possible," the authors could contextualize these issues as valuable directions for future research. Highlighting how the current findings serve as a foundation for addressing these more complex questions would not only strengthen the authors’ argument but also demonstrate an understanding of the broader scientific landscape.
In several instances, the responses directly address the reviewer’s technical concerns, such as issues related to methodology or data interpretation. For example, clarifications regarding parasitic capacitance and isothermal conditions provide valuable technical details. However, broader questions about methodological rigor, such as reproducibility statistics or the potential influence of experimental setups, are sometimes dismissed with phrases like "clear for experimentalists." Instead of assuming these points are self-evident, the authors could offer more thorough explanations or explicitly state how these concerns have been addressed in the manuscript. This would demonstrate attention to detail and a willingness to engage with the reviewer’s suggestions.
The reviewer’s comments on the Results and Discussion sections often focus on deeper mechanistic insights or comparisons to other systems. While the authors are correct to point out that addressing such questions would require additional research beyond the paper’s current scope, they could better emphasize how their findings contribute to understanding these broader issues. Linking the observed phenomena to potential theoretical implications or suggesting specific follow-up studies would provide a more constructive response and showcase the importance of the current work as a stepping stone for future investigations.
Similarly, the reviewer’s critique of the Conclusions section as lacking depth and broader implications was met with apparent frustration in the authors’ response. This critique, however, provides an opportunity to expand the conclusions slightly by highlighting potential applications or theoretical impacts of the findings, even in speculative terms. This approach would align the conclusions with the reviewer’s expectations without requiring significant additional analysis.
Overall, the responses could benefit from a more consistent acknowledgment of the reviewer’s valid points. Even when certain questions or suggestions fall beyond the immediate scope of the study, recognizing their significance and relevance to the field would demonstrate an openness to constructive criticism. By linking these questions to potential future work, the authors could turn perceived limitations into opportunities for further exploration.
In summary, the responses address many of the reviewer’s comments adequately in terms of technical detail, but the defensive tone and lack of engagement with broader implications undermine their effectiveness. A more neutral and professional tone, coupled with a constructive framing of scope-related limitations and a forward-looking approach to unanswered questions, would significantly improve the quality of the rebuttal. This would not only strengthen the authors’ position but also foster a more productive dialogue with the reviewer.
Author Response
Comments of the Reviewer 2 - comment on the response to Comments of Reviewer #1.
It has been rewritten and essentially softened. We had essential problem with the very extensive opinion of Reviewer #1.
The opinion shows the interest of the reviewer in the topic and presented research/
However, there is a set of statement highly inappropriate and beyond the scope of the report. There is a set for comments impossible to address, because the Reviewer suggests actions beyond the current state of the art. In few places he indicates the lack of issues which are explicitly given in the report.
We considered everything that was possible based on the given opinion in the report.
However, many comments are well beyond the scope and the target of the presented research. In fact, some suggestions mean the solution to existing puzzling questions for ODIC -formers, discontinuous transitions, and barocaloric effect. It is not possible in the given report. I am a specialist in the given field - working from decades in dielectric spectroscopy, high pressures and soft matte (including plastic crystals) and I was very surprised by some statements and suggestions of Reviewer # 1
It caused a rising irritation. I should calm dowwm and simply answer fact - by - fact, without and 'emotional indicators'. I hope that the corrected response to the opinion of Reviewer #1 is just like this.
At the beginning of the response, I also formulated the opinion that the enthusiasm of Reviewer #1, with the experience and superior equipment of my Lab, could realize the research program sketched by the Reviewer. For me, the opinion of the Reviewer is rather the research program for a few years, lasting projects that solve existing general problems - not the explicit opinion.
Prof. dr hab. Sylwester J. Rzoska
IHPP PAS, Warsaw, Polansd

Round 3
Reviewer 1 Report
Comments and Suggestions for Authors
The authors answered clearly the reviewer’s concerns.
New interesting applications and future works can be envisioned.